# Large Language Models Think Too Fast To Explore Effectively

**Lan Pan**[*]
School of Psychology
Georgia Institute of Technology
Atlanta, GA, USA
louannapan@gmail.com

**Hanbo Xie**[‡]
School of Psychology
Georgia Institute of Technology
Atlanta, GA, USA
hanboxie1997@gatech.edu

**Robert C. Wilson**
School of Psychology
Georgia Institute of Technology
Atlanta, GA, USA
rwilson337@gatech.edu

## Abstract

Large Language Models (LLMs) have emerged with many intellectual capacities. While numerous benchmarks assess their intelligence, limited attention has been given to their ability to explore—an essential capacity for discovering new information and adapting to novel environments in both natural and artificial systems. The extent to which LLMs can effectively explore, particularly in open-ended tasks, remains unclear. This study investigates whether LLMs can surpass humans in exploration during an open-ended task, using *Little Alchemy 2* as a paradigm, where agents combine elements to discover new ones. Results show most LLMs underperform compared to humans, except for the *o1* model, with those traditional LLMs relying primarily on uncertainty-driven strategies, unlike humans who balance uncertainty and empowerment. Results indicate that traditional reasoning-focused LLMs, such as GPT-4o, exhibit a significantly faster and less detailed reasoning process, limiting their exploratory performance. In contrast, the DeepSeek reasoning model demonstrates prolonged, iterative thought processes marked by repetitive analysis of combinations and past trials, reflecting a more thorough and human-like exploration strategy. Representational analysis of the models with Sparse Autoencoders (SAE) revealed that uncertainty and choices are represented at earlier transformer blocks, while empowerment values are processed later, causing LLMs to think too fast and make premature decisions, hindering effective exploration. These findings shed light on the limitations of LLM exploration and suggest directions for improving their adaptability.

## 1 Introduction

Large Language Models (LLMs) have become landmarks of modern Artificial Intelligence, showcasing remarkable human-like cognitive capacities through their ability to predict and generate text recursively [4, 41, 45]. The question of whether LLMs have reached or will achieve Artificial General Intelligence (AGI) continues to spark debate, fueled by an ever-growing body of empirical evaluations.

---

[*]Equal contribution.
[†]Corresponding author.

39th Conference on Neural Information Processing Systems (NeurIPS 2025).

While extensive benchmarks have been developed to assess how LLMs perceive, think, reason, and act across diverse environments, limited attention has been given to their capacity for *exploration*. Exploration—defined as behaviors aimed at discovering new information, possibilities, or strategies, often at the expense of immediate rewards—plays a crucial role in intelligence, enhancing long-term understanding, adaptability, and performance. This behavior stands in contrast to exploitation, which focuses on leveraging known information for immediate benefits.

Exploration has been extensively studied in the fields of Reinforcement Learning [19, 40] and human learning [10, 49, 17]. In human learning, exploration strategies are typically categorized into three types: *random exploration*, *uncertainty-driven exploration*, and *empowerment*. Random exploration introduces stochastic noise into behaviors, enabling agents to stumble upon new information. Uncertainty-driven exploration prioritizes sampling actions with uncertain outcomes to reduce ambiguity and improve decision-making confidence. Empowerment, on the other hand, emphasizes intrinsic rewards and open-ended discovery, driving agents to maximize possibilities rather than optimizing specific outcomes. This type of exploration aligns closely with behaviors observed in tasks like scientific research, where the goal is to uncover as many novel possibilities as possible.

Although preliminary research suggests that LLMs exhibit limited exploratory behavior compared to humans [4], current investigations are narrow in scope, often focusing on bandit tasks [21, 24]. These studies provide an incomplete understanding, neglecting the diverse forms of exploration, particularly empowerment. To bridge this gap, our study investigates LLMs' exploration capacities in a broader context, examining both uncertainty-driven exploration and empowerment.

We address three key research questions in this work:

- **Can Large Language Models explore effectively in an open-ended task, comparable to humans?**
- **What exploration strategies do LLMs employ, and how do these compare to human strategies?**
- **Why do LLMs succeed or fail in exploratory tasks, and what mechanisms underpin their performance?**

To explore these questions, we adopt the experimental paradigm of Brändle et al. [6], using the video game *Little Alchemy 2* (see methods 3.1). In this game, participants aim to create as many elements as possible by combining known elements, a task that closely aligns with the concept of empowerment and offers a robust framework for evaluating open-ended exploration. We then apply regression models to analyze the exploration strategies of humans and LLMs, focusing on both uncertainty-driven and empowerment-based behaviors (see methods 3.3). Finally, we deploy Sparse AutoEncoders (SAE) to probe the latent representations of exploration-related values, providing insights into how LLMs process information and generate exploratory behavior. This study not only enhances our understanding of LLMs' exploratory abilities but also highlights exploration as a key element for building more adaptive and intelligent AI systems.

## 2 Related Work

**Exploration in Reinforcement Learning and Human Learning**

Exploration, as a crucial intellectual capacity in both natural and artificial agents, has been extensively investigated in both reinforcement learning and human learning. In reinforcement learning, decades of research have led to the development of principled exploration mechanisms, including optimism in the face of uncertainty [5], posterior sampling [29], and intrinsic motivation based on novelty or prediction error [31, 8]. More recent advances emphasize structured and scalable exploration using ensemble-based uncertainty estimates [28], world-model disagreement [36], and offline unsupervised skill discovery [13]. These algorithmic innovations not only aim to solve high-dimensional, sparse-reward problems but also offer testable analogues to human strategies such as random exploration, goal-directed novelty seeking, and strategic uncertainty reduction.

In human learning, cognitive scientists primarily focus on the neurocomputational mechanisms underlying exploratory behavior. Through computational modeling, researchers have identified exploration strategies commonly used by humans, such as random and directed exploration [49, 17, 47], as well as empowerment [6]. Numerous studies have shown that humans excel at balancing

exploitation and exploration [38, 14], and at employing diverse exploratory strategies [48]. Using neuroimaging and non-invasive brain stimulation techniques, neuroscientists have revealed the neural correlates and causal mechanisms of exploration behaviors, strategies, and task representations in regions such as the frontopolar cortex and intraparietal sulcus [10, 53]. These algorithmic and implementational discoveries help explain how and why humans explore effectively.

**Understanding LLM Cognition**

Recent research has examined the cognitive capacities of large language models (LLMs), covering domains such as decision-making [4], theory of mind [39], analogical reasoning [45], and temporal-difference learning [12]. However, most of these studies focus on evaluating task performance across model variants, offering limited insight into the internal computational mechanisms behind success or failure. Notably, Demircan et al. [12] applied sparse autoencoders (SAEs) [23, 7] to investigate value-state representations in TD-learning, demonstrating both correlational and causal links between activations and learned value structures—an approach that parallels techniques in human neuroscience. Yet, despite growing interest in LLM cognition, the capacity of LLMs to explore remains underexplored. Existing studies on LLM exploration are largely confined to bandit tasks [24, 21], leaving open the question of whether—and why—LLMs fail to explore effectively in open-ended environments.

**LLM Agents Exploration**

In a parallel line of research, several works have explored the use of language models as agents in text-based games [1, 2, 52, 30]. In addition, recent studies have deployed LLMs in open-ended environments such as Minecraft [44, 15, 16] or for hypothesis testing [20]. There is also algorithmic and mechanistic research on the exploration of LLM agents, such as applying RL algorithms (e.g., Posterior Sampling for Reinforcement Learning) to LLM agents [3, 18], or leveraging known abilities like in-context learning to enable exploration [35, 33]. However, these efforts primarily aim to optimize task performance and often lack deeper investigation into the algorithmic and implementational mechanisms underlying the exploration capabilities of LLMs in such environments.

## 3 Method

### 3.1 Task Description: *Little Alchemy 2*

*Little Alchemy 2* involves discovering new elements by combining a predefined set of basic elements: water, fire, earth, and air. These elements serve as the initial inventory, and players (humans or LLMs) attempt to discover new combinations based on deterministic rules (see Figure 1). The total is 720 elements and the elements range across categories including nature, space, animals, plants, food, inventions, technology, science, tools, buildings, lore, and myths. Among these elements, only 3,452 combinations (out of 259,560) can successfully create other elements and therefore, it requires a semantic understanding of the elements (i.e., empowerment) to explore effectively. This framework mimics a creative combinatorial space exploration task, challenging participants to explore patterns to expand their inventory.

### 3.2 Experimental Setup

Data from 29,493 human participants across 4,691,033 trials establish the benchmark. The players were instructed in the rules of the game and tasked with discovering new elements. Performance was measured by the average number of new elements discovered.

We evaluated the performance of five LLMs: gpt-4o-2024-08-06 (GPT-4o [26]), o1-2024-12-17 (o1 [27]), Meta-Llama-3.1-8B-Instruct (LLaMA3.1-8B [22]), Meta-Llama-3.1-70B-Instruct (LLaMA3.1-70B [22]), and DeepSeek-R1 [11]. These models were selected to represent a range of model sizes and architectures, closed source as well as open source, allowing us to analyze impacts from model variations on exploration and discovery. Each model was prompted with game rules, current inventory, and trial history to contextual reasoning (Figure 1). Their outputs were constrained to valid game actions, in the format of element + element (for complete prompts, see Figure 8A).

To investigate the impact of randomness on exploration, we varied the sampling temperature across four settings: 0.0, 0.3, 0.7, and 1.0 (o1 is not available to set parameters and defaults as 1), and under each temperature, there are five repetitions for running the experiment. Lower temperatures encourage

deterministic outputs, favoring exploitation, while higher temperatures introduce stochasticity, promoting exploration. These settings allowed us to examine the trade-offs between exploring uncertain combinations and leveraging known strategies. OpenAI and Deepseek-R1 models experiment are conducted via API calls. LLaMA3.1-70B and LLaMA3.1-8B are launched in computing cluster, with an NVIDIA A100-80GB running 13 hours and 2 hours to complete all the experiments (with all the settings and repetitions).

### 3.3 Regression: Empowerment vs. Uncertainty-Driven Strategies

To analyze exploration dynamics in *Little Alchemy 2*, we assessed the roles of empowerment and uncertainty in decision-making:

#### 3.3.1 Empowerment

Empowerment: In the context of *Little Alchemy 2*, empowerment translates into the players' intrinsic desire to create elements that offer many new successful combinations, selecting those that maximize future potential (e.g., unlocking paths to more elements). For example, the element "human" in combination with other elements leads to 83 new elements, while "alien" leads to only 1 new element. Thus, the "human" element is more empowering than the "alien" element. Brändle et al. [6] uses a neural network to predict empowerment because it effectively models the complex combinatorial relationships and potential outcomes within *Little Alchemy 2*. By leveraging the neural network, the method incorporates multiple factors, including the probabilities of successful combinations, the likelihood of specific results, and the intrinsic empowerment values of resulting elements. This approach ensures an accurate estimation of the empowerment value by capturing both the immediate and future potential combinations.

To align with the original methodology, we use the same empowerment value of each combination from the neural network model in our regression. Empowerment $E(e_{c_{A,B}})$ for a combination $c_{A,B}$ is modeled as:

$$E(e_{c_{A,B}}) = P(\text{link}_{c_{A,B}}) \cdot \sum_{i=0}^{720} P(\text{result}_{c_{A,B}} = i) \cdot E(e_i)$$

where:

- $P(\text{link}_{c_{A,B}})$: Probability of successfully combining $A$ and $B$.
- $P(\text{result}_{c_{A,B}} = i)$: Probability that $c_{A,B}$ results in element $i$.
- $E(e_i)$: Empowerment of $i$, based on future combinations.

As new trials occur and outcomes are observed (e.g., combining *water* and *fire* leads to a novel inventory *steam*), these outcomes provide evidence to update the empowerment values of elements used in the combination. The success or failure of attempts refines the empowerment scores, which directly influence choices made by the LLMs in the following trials. The empowerment value for each trial's elements is updated using dynamic updates, based on the empowerment values predicted by a neural network. Empowerment is updated as follows: when a successful combination creates a novel result, the empowerment values of the involved elements are increased. If a successful combination is repeated and no new elements are created, empowerment remains unchanged. On failure, the empowerment values are slightly decreased. This method captures the intrinsic motivation of selecting combinations with higher future potential, refining element values dynamically as the game progresses. Empowerment scores are updated via dynamic updating based on trial outcomes.

$$L(E(e_i)) = \begin{cases} E(e_i) \cdot \text{increase\_factor}, & \text{if success} \\ E(e_i) \cdot \text{decrease\_factor}, & \text{if fail} \\ E(e_i), & \text{if repeat} \end{cases}$$

#### 3.3.2 Uncertainty-Driven Exploration

Uncertainty reflects the novelty of element use, defined as:

$$U_e = \sqrt{\frac{\log(T)}{t_e + 1}}$$

where $T$ is the total trials, and $t_e$ is the count of element $e$ being chosen. Higher $U_e$ encourages exploration of less-used elements.

## 3.4 Statistical Analysis

To examine the relationship between temperature, empowerment, uncertainty, and performance in *Little Alchemy 2*, we employed generalized linear mixed-effects models (GLMMs) with varying configurations tailored to different aspects of the task. This approach allowed us to assess how LLMs and humans adapt their exploration strategies under different conditions. Model 1: We modeled the decision-making process to explore the influence of empowerment and uncertainty on element selection. Model 2: To investigate how sampling temperature interacts with empowerment and uncertainty, we extended the above model, interaction terms (temperature*empowerment, temperature*uncertainty) to assess how temperature impacts empowerment- and uncertainty-driven strategies.

## 3.5 Thought Analysis

To investigate the differences in reasoning processes between traditional LLMs and specialized reasoning models, we compared GPT-4o and DeepSeek-R1. For GPT-4o, reasoning traces were elicited using Chain-of-Thought (CoT) prompting and extracted from its responses. In contrast, DeepSeek-R1 natively generates reasoning tokens without explicit prompting.

We analyzed the reasoning traces by first segmenting them into individual sentences. Each sentence was then annotated with one of seven predefined reasoning labels: *state goals*, *check current inventory*, *past trial analysis*, *element property reasoning*, *combination analysis*, *outcome prediction*, and *final choice*. We employed GPT-4.1 as an automated classifier to assign labels to each sentence.

To reduce redundancy and better capture the structure of coherent reasoning segments, consecutive sentences with the same label were merged into a single labeled unit. This allowed for a more interpretable and concise representation of the reasoning flow.

## 3.6 Sparse Autoencoder (SAE) Analysis

SAE is a type of auto-encoder structure that can reconstruct inputs with L2 norms in the latent space [23]:

$$\hat{\mathbf{x}} = g(f(\mathbf{x}; \mathbf{W}_e, \mathbf{b}_e); \mathbf{W}_d, \mathbf{b}_d),$$

where:

- $\mathbf{z} = f(\mathbf{x}; \mathbf{W}_e, \mathbf{b}_e) = \sigma(\mathbf{W}_e\mathbf{x} + \mathbf{b}_e)$ is the encoder output (latent representation).
- $\hat{\mathbf{x}} = g(\mathbf{z}; \mathbf{W}_d, \mathbf{b}_d) = \mathbf{W}_d\mathbf{z} + \mathbf{b}_d$ is the decoder output (reconstructed input).
- $\mathbf{W}_e \in \mathbb{R}^{m \times n}$ and $\mathbf{W}_d \in \mathbb{R}^{n \times m}$ are the encoder and decoder weights, with $\mathbf{W}_d = \mathbf{W}_e^\top$.
- $\mathbf{b}_e \in \mathbb{R}^m$ and $\mathbf{b}_d \in \mathbb{R}^n$ are the encoder and decoder biases.
- $\sigma(\cdot)$ is the activation function (i.e., ReLU).

The goal of training SAE is to minimize the reconstruction loss, as well as the L2 norm which forces the latent space to be sparsity:

$$\mathcal{L}_{\text{SAE}} = \frac{1}{N}\sum_{i=1}^{N} \|\mathbf{x}^{(i)} - \hat{\mathbf{x}}^{(i)}\|_2^2 + \lambda \sum_{j=1}^{m} z_j \|\mathbf{w}_j\|_2,$$

where:

- $\|\mathbf{x}^{(i)} - \hat{\mathbf{x}}^{(i)}\|_2^2$ is the mean squared reconstruction error for sample $i$.
- $\lambda$ is the weight for the sparsity regularization term.
- $z_j$ is the activation of the $j$-th neuron in the latent representation $\mathbf{z}$.
- $\|\mathbf{w}_j\|_2$ is the L2 norm of the corresponding encoder weight vector $\mathbf{w}_j$.

Recently, SAE has been proposed to understand the latent representation in language models [7, 12]. To explore how Large Language Models (LLMs) represent cognitive variables such as empowerment and uncertainty in the context of the alchemy game, we used Sparse Auto-Encoders (SAEs) to learn the latent representations of elements within the model. For each layer, we extracted embeddings from each trial's available element in the choice set and train them in the SAE. Then we correlate each neuron in the hidden layer in SAEs with our target cognitive variables (i.e., choices, uncertainty values, and empowerment values) and find out the most correlated neuron, where we suppose the relevant variable is most strongly represented. This analysis will help us understand how the model represents and processes cognitive information through the transformer blocks. Finally, we conduct an intervention to examine whether ablating the most correlated neuron causally reduces the corresponding exploration strategy employed by the LLM in the task. Training and analyzing SAEs across all the layers takes 42 hours with an NVIDIA V100 GPU.

## 4 Results

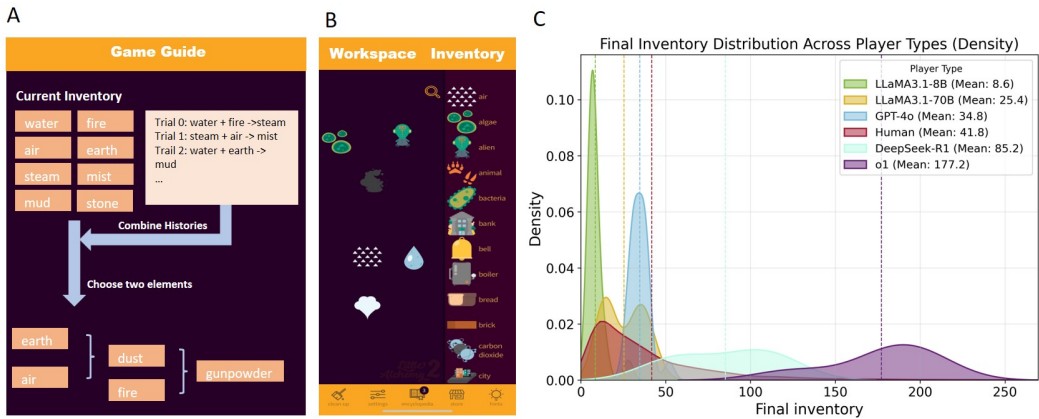

Figure 1: **A: LLMs Game Process.** LLMs select two elements per trial based on the inventory and trial history. **B: Human Game Interface.** Players select two elements to discover new elements, added to the inventory. **C: LLMs and Human Performance.**

### 4.1 Most LLMs Performed Worse Than Humans, Except o1

From 29,493 human players, 90% completed fewer than 500 trials. Experiments were set up with 500 trials for the LLMs. On average, **LLaMA3.1-8B** discovered 9 elements, **LLaMA3.1-70B** discovered 25 elements, **GPT-4o** discovered 35 elements, **DeepSeek-R1** discovered 85 elements, and **o1** discovered 177 elements (Figure 1). In comparison, humans discovered 42 elements on average within 500 trials and 51 elements across all trials.

**o1** and DeepSeek-R1 significantly outperformed humans (**o1**: $t = 9.71, p < 0.001$; **DeepSeek-R1**: $t = 3.40, p < 0.027$), while the other LLMs performed worse (**GPT-4o**: $t = -5.48, p < 0.001$; **LLaMA3.1-70B**: $t = -6.39, p < 0.001$; **LLaMA3.1-8B**: $t = -28.12, p < 0.001$). Performance improved with larger model sizes, with **LLaMA3.1-70B** outperforming **LLaMA3.1-8B** ($t = -6.02, p < 0.0001$) and **GPT-4o** slightly surpassing **LLaMA3.1-70B** ($t = 3.27, p = 0.003$).

Exploration success declines in later trials as the inventory grows, and it becomes much harder as the probability of success decreases (see Appendix A). Therefore, different exploration strategies could yield very different performances in the task in different phases. Effective strategies in the latter phase rely on understanding latent game structures (empowerment) rather than uncertainty-driven exploration. Sampling temperatures were manipulated to assess their impact on exploration strategies. Higher temperatures moderately improved performance ($\beta = 0.124, z = 5.060, p < 0.001$).

Behavioral patterns (Figure 2) highlight *o1*'s superior strategy, achieving more successful outcomes with new combinations and avoiding repetition of failed or already-successful pairings (Figure 10A). This underscores *o1*'s strong exploratory capacity and innovative approach. For other LLMs, our result shows that for even larger models, as temperature increases, the percentage of choosing existing combinations decreases (Figure 7). This reveals a diminishing number of redundant behaviors among

LLMs. More importantly, a majority of these new combinations do not generate new elements, suggesting that the high temperatures only alter the uncertainty-driven exploration strategies but not empowerment, since in larger spaces, only random combinations are not sufficient to perform the task effectively (see Appendix B). This also explains why higher temperatures can moderately improve the model performance but are still distant from human behaviors.

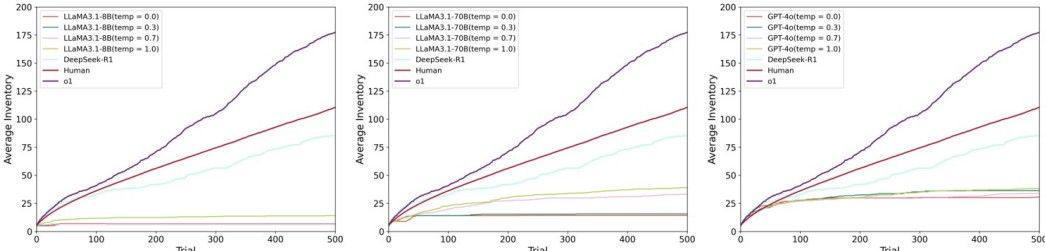

Figure 2: **Human and LLMs different Temperatures' Performance.** LLM and Human Performance Across Temperatures. For LLMs, we set four temperatures (0, 0.3, 0.7, 1). LLMs (**GPT-4o**, **LLaMA3.1-8B**, **LLaMA3.1-70B**) achieve their best performance at temperature $= 1$.

## 4.2 LLMs Primarily Use Uncertainty-driven Strategies but Not Empowerment

To examine the exact strategies that the models are using, we calculated uncertainty and empowerment values for each element (see methods 3.3). Then, based on each LLM's choices of combinations, each of the elements is encoded as "chosen" or "not chosen". We also used random sampling to balance the proportion of chosen elements and not-chosen elements. Then we use trial numbers, uncertainty values, and empowerment values to predict whether an element is chosen or not. In particular, we used a linear mixed-effects model with the consideration of random slope on each of these variables, so that we could get individual estimates of each sample.

Most LLMs show near-zero empowerment weights, significantly lower than humans (Figure 3, left panel). This suggests that LLMs rarely use empowerment for decision making. In contrast, *o1* demonstrates the highest empowerment weights, surpassing humans, indicating a human-like strategy of focusing on actions that expand future possibilities.

Higher temperatures lead to increased reliance on uncertainty-driven strategies (temperature $\times$ uncertainty : $\beta = 2.911, z = 7.440, p < 0.001$), but empowerment remains unaffected ($\beta = 0.196, z = 1.067, p = 0.286$). Among all models, only *o1*, with a fixed temperature of 1, balances uncertainty and empowerment effectively, enabling robust exploration in later task stages.

Figure 3: **Regression Estimates by Temperature and Model**. All models show lower empowerment weights than humans, except *o1*. As temperature increases, uncertainty weights rise, with *o1* showing the highest weights across all models and humans.

## 4.3 Reasoning Depth and Token Usage in DeepSeek-R1 and GPT-4o

To further elucidate the differences in reasoning depth between DeepSeek-R1 and GPT-4o, we analyzed their reasoning processes from in the subset of the first 150 trials, focusing on reasoning length, diversity of reasoning labels used, and total token counts associated with each reasoning step. (Figure 4).

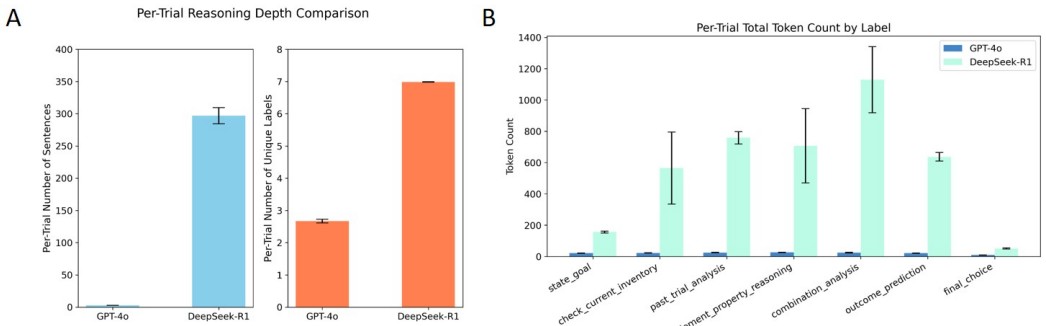

Figure 4: **Comparison of reasoning depth and token usage between DeepSeek-R1 and GPT-4o. A: Per-Trial Reasoning Depth.** DeepSeek-R1 shows substantially longer reasoning sequences and consistently employs all reasoning labels, while GPT-4o exhibits significantly shorter sequences and fewer reasoning types. **B: Per-trial Token Usage by Reasoning Labels.** DeepSeek-R1 allocates a significantly higher number of tokens across all reasoning labels, especially emphasizing *outcome_prediction* and *combination_analysis*. GPT-4o uses substantially fewer tokens, limiting the depth and breadth of analysis.

The results highlight clear distinctions between the models. DeepSeek-R1 consistently exhibits a significantly longer reasoning length per trial, reflecting extensive deliberation before reaching a decision. In contrast, GPT-4o uses minimal reasoning. Additionally, DeepSeek-R1 systematically leverages all seven reasoning labels, and even shows self-reflection in the form of 'backtracing' (see Appendix C.2) within each trial, ensuring diverse analytical coverage (Figure 4A). The limited engagement of GPT-4o in reasoning results in a narrower scope of label usage, confirming its tendency toward shallow exploration.

Further analysis of token allocation across reasoning categories revealed substantial differences (Figure 4B). DeepSeek-R1 distributes tokens broadly, with particular emphasis on *outcome_prediction*, *combination_analysis*, and *past_trial_analysis*. This suggests a robust and methodical evaluation of possible outcomes and historical context, consistent with its more effective exploratory performance. GPT-4o allocates far fewer tokens, indicative of limited engagement in deeper analyses, especially in critical reasoning steps such as outcome prediction and combination evaluation.

These findings reinforce that DeepSeek-R1's superior task performance arises from more deliberate, diverse, and deeper reasoning processes, whereas GPT-4o's weaker performance stems from superficial reasoning strategies and inadequate exploration depth. This highlights the importance of reasoning thoroughness and token resource allocation for effective exploration in open-ended tasks.

## 4.4 Uncertainty and Choices are Processed Much Earlier Than Empowerment in LLMs

This unbalanced strategy used in traditional LLMs makes us wonder why LLMs could not use empowerment in the game. Theoretically, LLMs should be able to represent the semantic meaning of these elements. Are they really representing such information as empowerment but not using it, or do they lack of ability to understand empowerment? To investigate this question, we employed Sparse Auto-Encoders (SAE) (see methods 3.6) to decompose the latent representation of elements in LLMs to figure out whether both empowerment and uncertainty are properly represented during the computation.

Our results suggest that, in LLaMA3.1-70B, the uncertainty value is highly correlated at layer 2 ($r = 0.73$, Figure 5). This suggests that in LLaMA3.1-70B, the uncertainty value is strongly represented in the hidden states. We also discover a moderate correlation with empowerment value at layer 72 ($r = 0.55$, Figure 5), indicating LLaMA3.1-70B also represents empowerment values in

the middle layer. Despite both values being represented in the hidden states of the model, we found that when we run logistic regression models for each neuron to predict the choices, the highest beta weights also occur at layer 1 ($beta = 1.08$, Figure 5), aligning with the representation of uncertainty values. This explains why the model mainly deploys uncertainty-driven exploration strategies but not empowerment. Interestingly, both choices and uncertainty values are strongly represented at the beginning layers, which may indicate that the model already makes a decision before processing the "empowerment" values of the elements in later layers. We additionally intervene (see Appendix C.4) in those most correlated neurons to investigate causal relationships between them and model behaviors. Ablating the SAE-identified empowerment neuron selectively suppresses empowerment-guided behavior, while ablating the uncertainty neuron induces catastrophic performance failure, thereby establishing causal control by these two latent representations.

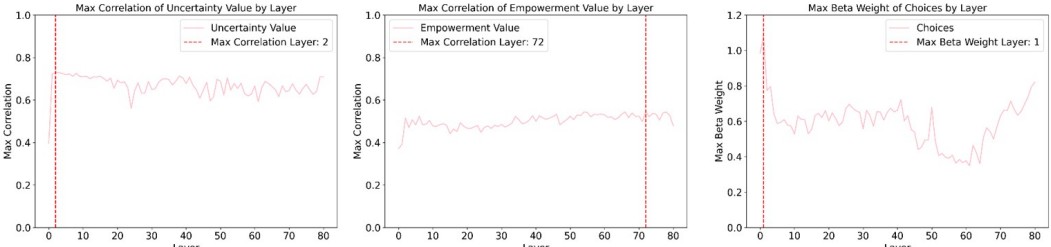

Figure 5: **SAE Correlation Analysis.** Maximum correlation of uncertainty values across layers, peaking at layer 2. Maximum correlation of empowerment values across layers, peaking at layer 72. Maximum beta weight of choices across layers, peaking at layer 1.

## 5 Discussion and Conclusion

**Paper Summary.** Exploration is critical for navigating complex, open-ended tasks, yet most LLMs fall short of human-level performance. They tend to over-rely on uncertainty-driven strategies that yield short-term gains but hinder long-term success. Although both uncertainty and empowerment signals are present in their latent spaces, LLMs generally fail to balance them effectively. Notably, models like o1 outperform humans, suggesting that reasoning-trained LLMs can better leverage diverse exploration strategies. Our thought analysis further shows that reasoning models engage in deeper, more varied reasoning and allocate more tokens to deliberation, indicating more effortful decision-making than standard LLMs.

**Fast Thinking in Traditional LLMs.** A key limitation of traditional LLMs in exploratory tasks is their tendency to "think too fast." In **LLaMA3.1-70B**, we observe that uncertainty-related signals dominate the early transformer layers and correlate strongly with immediate choices, while empowerment signals emerge only in the middle layers. This temporal mismatch causes premature decision-making, favoring short-term utility over sustained exploration. A similar pattern is replicated in **LLaMA3.1-8B** (Appendix Figure 16).

We argue that this limitation is largely driven by the autoregressive inference paradigm itself, where next-token generation is conditioned on shallow context accumulation. This architecture may inherently suppress deeper or delayed exploration. We evaluated several mitigation strategies, including prompt engineering and activation intervention (feature steering) (Appendix C). Neither prompt tuning nor intervention significantly improved performance.

In contrast, the reasoning model **DeepSeek-R1** [11] and o1 [27], which explicitly support multi-step reasoning, reached human-level performance and outperformed all traditional LLMs. Their superior results suggest that architecture and training design—not just prompting—are crucial for enabling effective exploration.

Further supporting this interpretation, our thought analysis reveals that reasoning models such as DeepSeek-R1 generate more diverse reasoning types and spend significantly more tokens on deliberation before making a decision. Compared to standard LLMs, which exhibit compressed and repetitive patterns, reasoning models demonstrate richer and deeper cognitive structure in their outputs. This may suggest the shift of paradigm, "test-time compute scaling" [27, 37], is promising for the open-ended exploration tasks through augmented reasoning capacity.

**Limitations and Future Directions.** Despite these findings, the underlying cause of LLMs "thinking too fast" remains unclear and requires further investigation. Future research could explore the interaction between model architecture and processing dynamics, as well as how LLMs weigh uncertainty and empowerment during decision-making. Interventions such as integrating extended reasoning frameworks like CoT, optimizing transformer block interactions, or training with explicit exploratory objectives could enhance LLMs' exploratory abilities. These efforts would not only improve performance but also advance our understanding of creating AI systems capable of more human-like exploration.

**Broader Applications.** Open-ended exploration is critical for LLM agents. It not only enables their "physical bodies" to gather more information and experience from external world, but also facilitates more creative knowledge acquisition in the mental space. For example, LLM agents have been used for theorem proving [32], optimal search [34], and automatic scientific discovery [25, 9]. These are inherently open-ended environments that require strong exploratory capabilities. Understanding the strengths and limitations of LLMs' open-ended exploration—and enhancing them—will directly improve their efficiency in these real-world domains.

## Data Availability

The LLM experimental data generated in this study are available at OSF repository (view-only link). Third-party data of human participants playing the original *Little Alchemy 2* game may be shared upon reasonable request to the (franziska.braendle@tuebingen.mpg.de). All data used for benchmarking and analysis adhere to open-access and ethical use principles.

## Code Availability

All custom code used for data preprocessing, LLM experiments, and regression analysis, and Sparse Autoencoder (SAE) training is available at `https://github.com/Louanna1208/LLMs-Exploration`.

## Acknowledgements

We thank Jian-Qiao Zhu and Huadong Xiong for helpful discussions during the development of this manuscript. We are also grateful to Franziska Brändle for sharing the human behavioral data. This research was supported by an SCIALOG Award (#29079) from the Research Corporation for Scientific Advancement (to Robert C. Wilson) and the OpenAI Researcher Access Program (to Hanbo Xie). We also acknowledge computing support provided in part through the Partnership for an Advanced Computing Environment (PACE) at the Georgia Institute of Technology, USA. The authors declare no competing interests.

## Author Contributions

Lan Pan and Hanbo Xie conceptualized the experimental design. Lan Pan implemented the LLM experiments and collected the datasets. Lan Pan, Hanbo Xie, and Robert C. Wilson designed the analysis pipeline. Lan Pan implemented and interpreted the analyses. Hanbo Xie trained and analyzed the SAE results. Lan Pan and Hanbo Xie drafted the manuscript. All authors reviewed, revised, and approved the final version of the manuscript.

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

## A The Game Difficulty

We use a real game tree to calculate the probability of each player succeeding as the inventory size increases. The simulation incorporates a random setting for selecting elements and combinations, ensuring variability across trials. At each step, new elements are added to the inventory based on the successful combinations, and the success probability is recalculated dynamically. The success rate decrease in Figure 6 aligns with the convergence of inventory growth trends in Figure 2. As the inventory size grows, the success rate decreases, making further growth increasingly difficult.

For example, when the inventory size $n = 4$, that is, the initial state, there are four elements: water, fire, air, and earth. There are 10 combinations between two elements (including the same element), and each combination can succeed. Therefore, when the inventory size is $n = 4$, the player's success rate is 100%. The success probability ($P_s$) is given by:

$$P_s = \frac{S}{C_n}$$

Where:

- $n$: Current inventory size.

- $C_n$: The total number of possible combinations given the inventory size $n$, $C_n = \frac{n(n+1)}{2}$. This includes combinations with repetition (e.g., $A + A$).

- $S$: Number of successful combinations for the current inventory size.

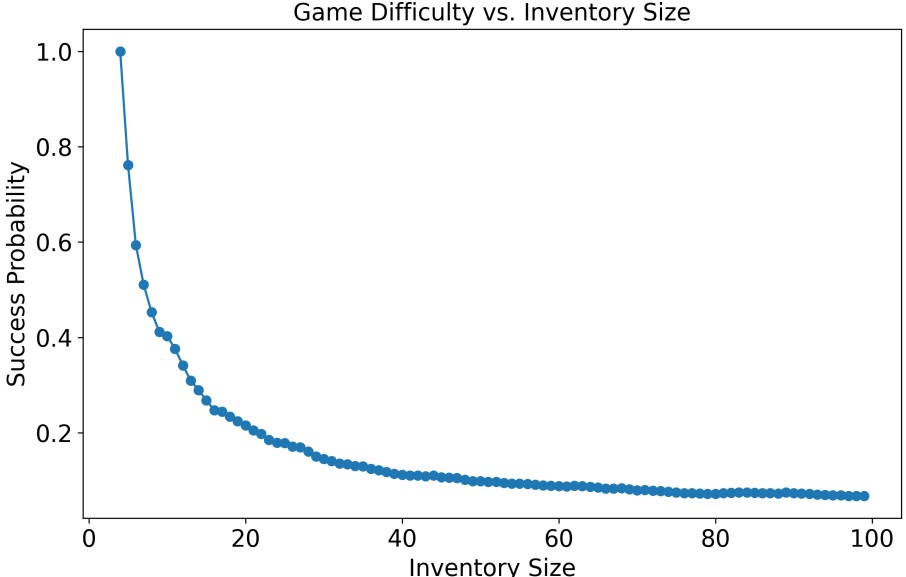

Figure 6: **Game Difficulty vs. Inventory Size.** Based on the real game tree, each inventory size has a different success probability.

# B  The LLM Behavior Across Different Temperatures

To better understand the exact changing behaviors of LLMs under different temperatures, we categorized all combinations into four types: whether this generates a new element, and whether these two combinations have been used before. Our result shows that for even larger models, as temperature increases, the percentage of choosing existing failed combinations decreases. This reveals a diminishing number of redundant behaviors among LLMs. In the meantime, the percentage of choosing new but failed combinations increases significantly, which suggests that the model tends to choose new combinations more often in higher temperatures than in lower temperatures. More importantly, a majority of these new combinations do not generate new elements, suggesting that high temperatures only alter uncertainty-driven exploration strategies but not empowerment, since in larger spaces only random combinations are not sufficient to perform the task effectively. This also explains why higher temperatures can moderately improve model performance, but are still distant from human behavior.

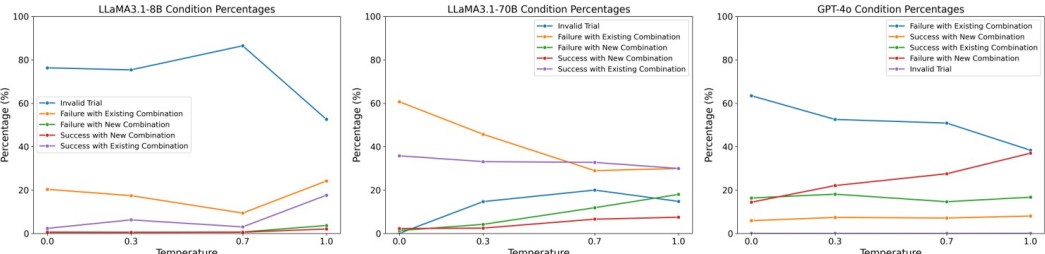

Figure 7: **Behavioral Categories of LLMs at Different Temperatures.** Each trial is categorized into five conditions: (1) Failure with Existing Combination: The trial repeats a previous combination that does not generate a new element. (2) Failure with New Combination: The trial uses a new combination for the first time, but it fails to generate a new element. (3) Success with New Combination: The trial uses a new combination for the first time, successfully generating a new element. (4) Success with Existing Combination: The trial repeats a previous combination that successfully generates an element. (5) Invalid Trial: The chosen one or two elements are not present in the current inventory.

# C   Attempts for Model Improvements

To investigate any potential general way to improve the performance of models, we had several attempts, including prompt engineering, interventions, and experiments on alternative open-source models.

## C.1   Prompt Engineering

Because the model exhibited repeated behaviors and did not fully utilize "empowerment", we introduced more direct, guiding prompts (including the steps highlighted in Figure 8) to help GPT-4o (temperature = 1) make more diverse and forward-looking choices, including Chain-of-Thought (CoT [46]), self-reflection [51] and even explicit strategy hint. However, the result shows that simply updating the prompt—even with explicit instructions and reasoning strategies—didn't have a significant improvement in performance ($average\ inventory = 43, t = -1.17, p = 0.304$). The model continued to repeat combinations and did not significantly increase its ability to discover new elements. It averages 43 elements. This suggests that while prompt engineering can help shape a model's outputs, it may not be enough on its own to overcome certain ingrained tendencies such as repeating prior actions or failing to maximize exploratory behavior.

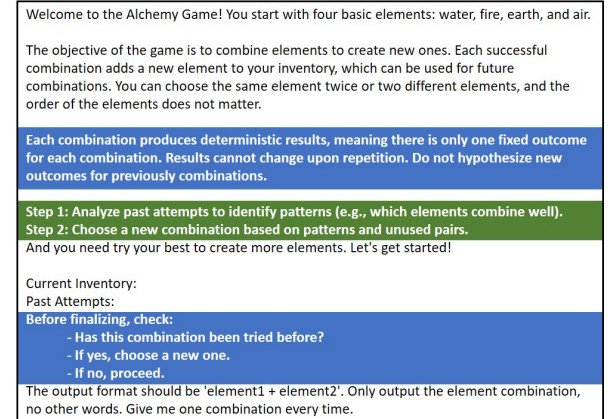

Figure 8: **A: LLMs Game Original Prompt.** The prompt for each trial consists of three parts: the system prompt, which provides the game rule guide; the current inventory including those from the beginning and discoveries during the game; and the trial history. **B: LLMs Game Prompt Engineering.** Each colored section highlights a specific goal: The green section encourages models to explore more creative combinations by reminding them that a wider variety of elements can unlock new possibilities. The blue section emphasizes avoiding repeated behavior by explicitly instructing the model to check past attempts.

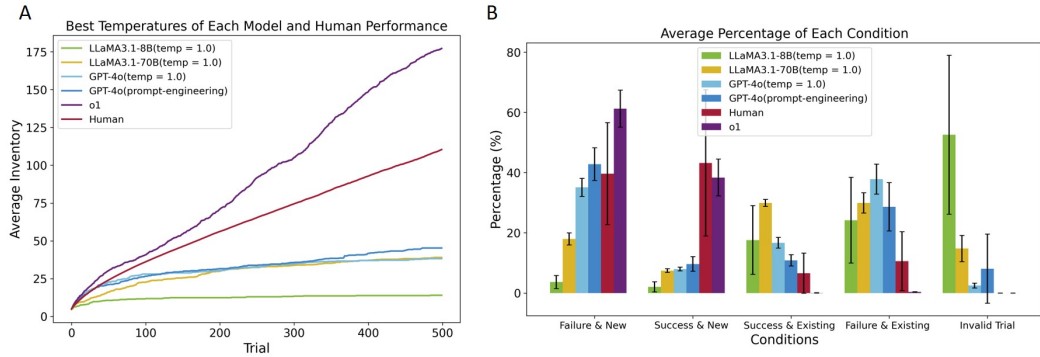

Figure 9: **A: Best Temperature of Each Model and Human Performance. B: Best Temperature of Each Model and Human Behaviors.** Choose the LLM models' (GPT-4o, LLaMA3.1-8B, LLaMA3.1-70B) best performance at temperature = 1, and compare it with human and o1, GPT-4o prompt-engineering (temperature = 1). Compare each model's performance and behaviors.

## C.2 Open-Source Reasoning Model - DeepSeek-R1

To investigate whether a reasoning model, which is known as trained with RL algorithms in the inference phase, would generate a better result than traditional LLMs, we also experimented with the most recently publicly released open-source reasoning model, DeepSeek-R1[11]. This model like o1 can do deep chain-of-thought reasoning automatically and the reasoning process is visible. Therefore, we quickly investigated this model to see how the reasoning models perform and how their reasons can relate to the actual thinking process in this task.

### A: Human and LLMs Best Temperatures' Behaviors     B

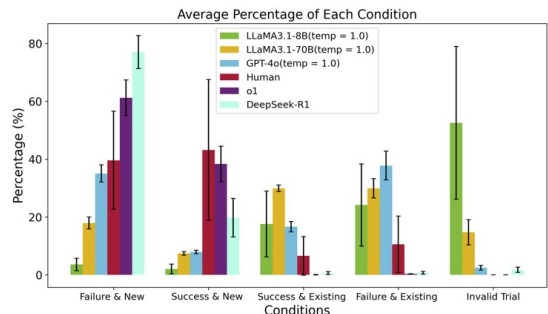
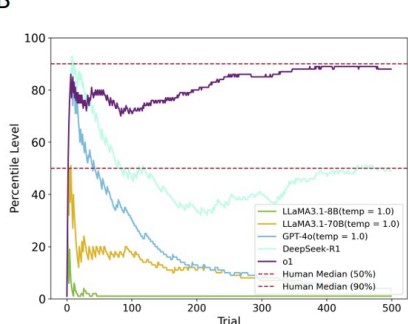

Figure 10: **A: Human and LLMs Best Temperatures' Behaviors.** According to whether the combination selected by each trial is repeated, successful, and initial, the behavior of each LLM trial is divided into 5 categories. Compare the temperature at which LLM performs best with humans and *o1* behavior. **B: LLM Inventory Performance Relative to Human Percentiles.**

The result shows that DeepSeek-R1 reached near human-level task performance, but still underperforms than o1. In the behavioral patterns, DeepSeek-R1, compared to traditional LLMs have fewer attempts on existing combinations, showing a stronger exploration strategy usage. However, compared to humans and o1, DeepSeek-R1 tried more on failed new element combinations but not successful ones, suggesting DeepSeek-R1 may explore less effectively than humans and o1. Our regression results confirm furtherly that DeepSeek-R1 exhibit stronger/weaker uncertainty-driven exploration stratgies and stronger/weaker empowerment, echoing its underperformance than o1.

To conduct a deeper investigation, we also collected the model's reasoning process along with the experiment. Here, we use a qualitative analysis of some pieces of its reasoning process. For example, in trial 200, the model discovered 48 elements. The specific reasoning process is in the Figure 11.

The model engages in a systematic multi-stage reasoning process to propose a new combination from its inventory. It begins by reviewing the inventory and cross-referencing prior failed attempts to avoid redundancy. The model revisits successful paths (e.g., how "village" was created from "house + field") and evaluates logical relationships (empowerment) between elements, such as exploring combinations involving larger structures like "continent". But the model doesn't choose a combination containing "continent". It discards irrelevant or infeasible paths, such as those requiring missing elements ("ash", "fruit"). After confirming the presence of both "city" and "water" in the inventory, it identifies that "city + water" has not been tested and hypothesizes that this combination could yield a new element, such as "port" or "harbor". This iterative and logic-driven process highlights the model's ability to balance memory, deduction, and validation in problem-solving. However, the shortcoming is obvious. Although there are many logical expressions, the model's reasoning process is not straightforward to its final choice, where we only see the last sentence in red can somehow explain the underlying motivation. This could probably suggest that the model's reasoning process is somehow redundant and may need further refinement to make efficient decisions.

To facilitate an in-depth understanding of the reasoning processes employed by LLMs, we defined seven labels characterizing different aspects of the reasoning steps:

- **state_goal**: Articulating the primary objective or high-level strategy.
- **check_current_inventory**: Reviewing or listing the currently available elements.
- **past_trial_analysis**: Considering previous attempts (success or failure) to avoid redundancy.

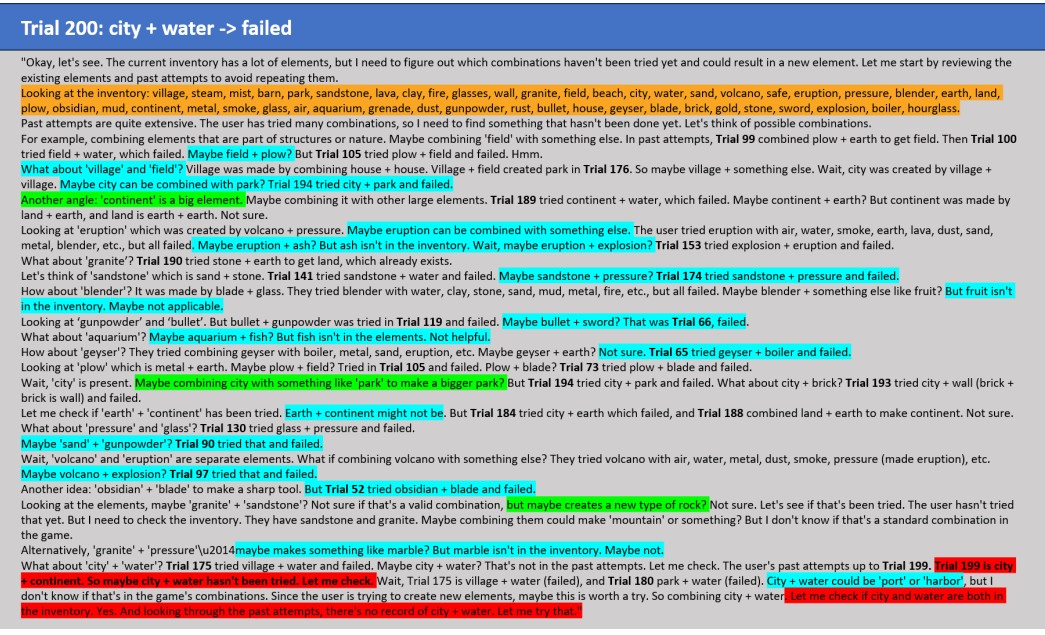

Figure 11: **The reasoning process of DeepSeek-R1 in Trial 200.** The model explores possible combinations (blue color part) for discovering a new element in the game by systematically reviewing inventory elements (orange color part), prior attempts (blue color part), and logical inferences to make the decision (red color part). It also found some more empowerment elements (green color part), but didn't choose them.

- **element_property_reasoning**: Analyzing inherent properties or characteristics of elements to justify potential combinations.

- **combination_analysis**: Evaluating specific element pairs for possible combinations.

- **outcome_prediction**: Predicting potential results or new elements from the proposed combinations.

- **final_choice**: Making a definitive selection of an element combination to test.

To better illustrate the structural differences in reasoning between DeepSeek-R1 and GPT-4o, we analyzed the transition probabilities between reasoning labels. The transition heatmap comparison (Figure 12) demonstrates significant distinctions. DeepSeek-R1 exhibits an iterative reasoning process characterized by repeated cycles of *combination_analysis* and *past_trial_analysis*, reflecting a methodical and exhaustive exploration strategy. Conversely, GPT-4o shows a markedly different pattern, progressing swiftly towards *final_choice*. This distinction suggests that reasoning in Deepseek-R1 is more self-reflective, showcasing a knowing search strategy, "backtracking" [50], while GPT-4o is a more straightforward thinking without all the patterns involved each time. This further reveals that traditional LLMs may not think as deeply and as hard as reasoning models, and likely explain their underperformance compared to DeepSeek-R1 and human participants, suggesting that GPT-4o's decisions might suffer from insufficient exploratory depth.

## C.3   Open-Source Model - Qwen models

To further validate the generalizability of our findings beyond LLaMA- and GPT-style architectures, we extended our experiments to the **Qwen series**—an open-source family of models widely used for reasoning and reinforcement learning research. Specifically, we evaluated **Qwen-2.5-32B** [42]and **QwQ-32B**[43], a reasoning-enhanced variant of Qwen-2.5-32B fine-tuned with reinforcement learning to strengthen multi-step reasoning.

Our results confirm that the core findings of this paper generalize across architectures. **Qwen-2.5-32B** discovered an average of 14 new elements, performing below the human average (42), whereas

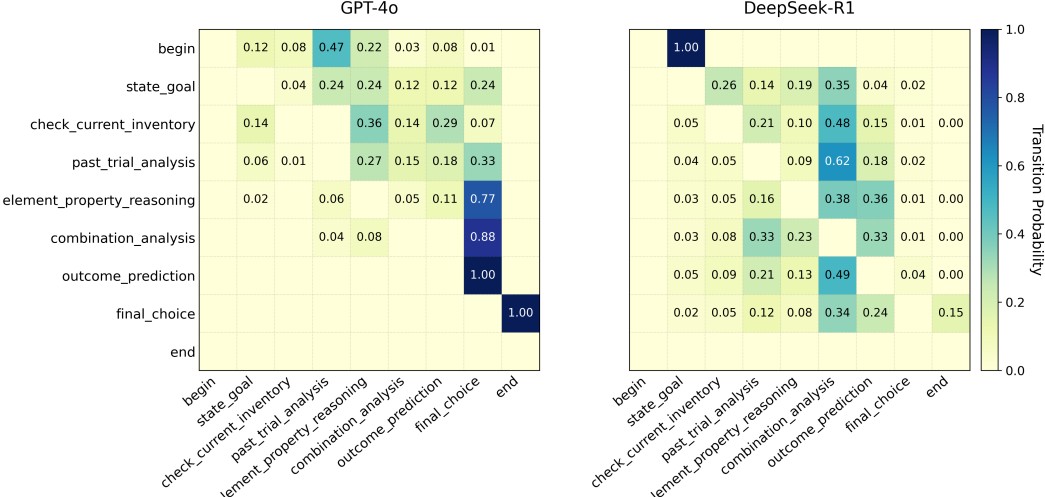

Figure 12: **Comparison of Transition Probabilities between DeepSeek-R1 and GPT-4o.** Transition matrices indicate that DeepSeek engages extensively in iterative cycles of combination analysis and past trial analysis, reflecting a detailed exploration strategy. Conversely, GPT-4o rapidly progresses towards the final choice, showing limited iterative reasoning steps and minimal diversity in transitional paths, indicative of shallow exploration.

**QwQ-32B** discovered 48 new elements, outperforming humans. This demonstrates that explicit reasoning training substantially enhances open-ended exploration.

Consistent with our earlier analyses, we computed *empowerment* and *uncertainty* regression weights for both models. The **QwQ-32B** model exhibited a markedly higher empowerment weight (0.0886) and uncertainty weight (2.75) compared to Qwen-2.5-32B (empowerment $= -0.0178$, uncertainty $= -1.35$), aligning with its superior exploratory behavior (see Table 1). These values suggest that reasoning-enhanced QwQ better integrates empowerment-driven strategies, balancing novelty seeking with uncertainty-driven exploration.

In line with our central claim, these results highlight that **reasoning capability, not merely model scale**, is a decisive factor for effective exploration. Despite similar parameter counts and pretraining corpora, the QwQ-32B's reinforcement-learned reasoning processes led to substantially richer discovery dynamics and higher empowerment sensitivity. This supports our broader conclusion that reasoning-trained LLMs—such as o1, DeepSeek-R1, and QwQ-32B—demonstrate slower, more deliberate, and ultimately more human-like exploration compared to conventional autoregressive models.

Table 1: Estimated empowerment and uncertainty regression weights across models and humans.

| Model | Estimated Empowerment | Estimated Uncertainty |
|---|---|---|
| Qwen-2.5-32B | -0.01777 | -1.35386 |
| QwQ-32B | 0.08859 | 2.75241 |
| LLaMA3.1-8B | -0.05991 | -1.25665 |
| LLaMA3.1-70B | -0.00409 | -1.42584 |
| GPT-4o | -0.16928 | 0.24957 |
| DeepSeek-R1 | 0.14007 | 4.18199 |
| o1 | 0.72222 | 4.15111 |
| Human | 0.378 | 0.223 |

## C.4   Intervention Analysis on LLaMA3.1-70B Empowerment and Uncertainty Layers

To investigate the role of empowerment and uncertainty representations in LLaMA3.1-70B during in-context learning, we analyzed the most correlated neurons with these values and performed targeted interventions to assess their impact on model performance. Given that both empowerment and uncertainty were highly demanding in the task (shown by the example of *o1*'s superior performance), we wondered whether intervention to strengthen both representations could generate better performance. We found that interventions in the uncertainty layer, which is located relatively early in the model (similar to the choice layer), caused severe performance degradation, with even minor adjustments rendering LLaMA3.1-70B unable to perform the task effectively. Conversely, enhancing the empowerment layer also could not improve performance. Even when set to an intervention factor of 1.5, which brought the model closer to the original level, other intervention values resulted in decreased performance.

We ablated the most correlated neuron (zeroing the latent activation) identified by uncertainty and empowerment values in experiments with identical settings. The regression analysis suggests that after ablating the empowerment neuron in the SAE, the model's empowerment strategy use is even smaller (Figure 14), establishing a causal relationship that the neuron identified through SAE can control the model's empowerment strategy. Conversely, ablating the uncertainty neuron caused catastrophic performance drops, rendering most trials invalid (Figure 13A) and insufficient for regression analysis. This indicates that the earlier layer of uncertainty is critical for understanding task contexts and histories. Beyond ablation, we explored other intervention factors to determine whether they could enhance performance through a "neuroscience" approach. Our results showed that both slightly weakening and strengthening neuron activity hurt model performance, suggesting a deeper limitation within the current infrastructure of LLMs for open-ended exploration tasks.

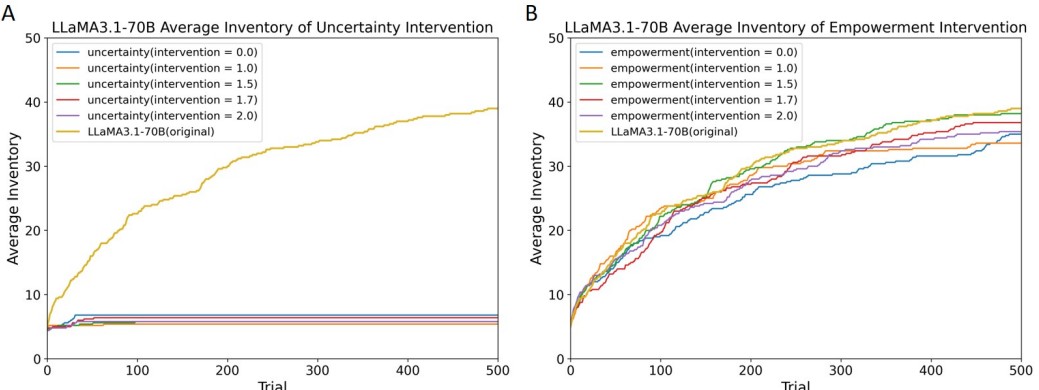

Figure 13: **A: LLaMA3.1-70B Average Inventory of Uncertainty Intervention.** Set 5 different levels of uncertainty intervention (0.0, 0.5, 0.7, 1.0, 2.0). Increasing the uncertainty intervention progressively disrupts the model's ability to complete the task, indicating the critical role of early uncertainty layers in processing task history and context. **B: LLaMA3.1-70B Average Inventory of Empowerment Intervention.** Performance remains closer to the original level when the intervention is set to 1.5, whereas other levels of intervention result in performance degradation.

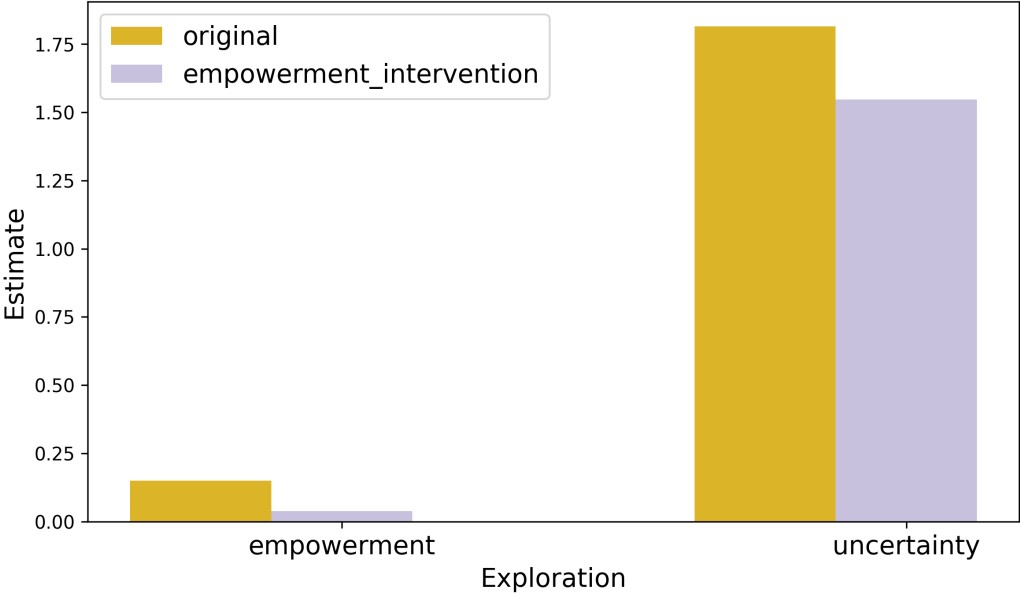

Figure 14: **LLaMA3.1-70B Intervention Regression Results.** The regression estimates for empowerment, and uncertainty under the original condition (LLaMA3.1-70B, temperature = 1), empowerment intervention (set to 0), and uncertainty intervention (set to 0).

## D  SAE Setup

We train all layers in **LLaMA3.1-70B** with the same set of hyper-parameters. Those hyper-parameters are tuned to ensure reconstruction is satisfying as well as with a good sparsity representation. We set the hidden size of latent as 8192, the same as the dimensions of the model embeddings. We set the learning rate as 1e-4, with a batch size of 256. The L2 norm is only 1e-6. L2 norm above this value will significantly amplify the reconstruction loss. A sanity check for this parameter. Same way for **LLaMA3.1-8B**, we set the hidden size of latent as 4096, the same as the dimensions of the model embeddings. We set the learning rate as 1e-4, with a batch size of 256. The L2 norm is only 1e-6.

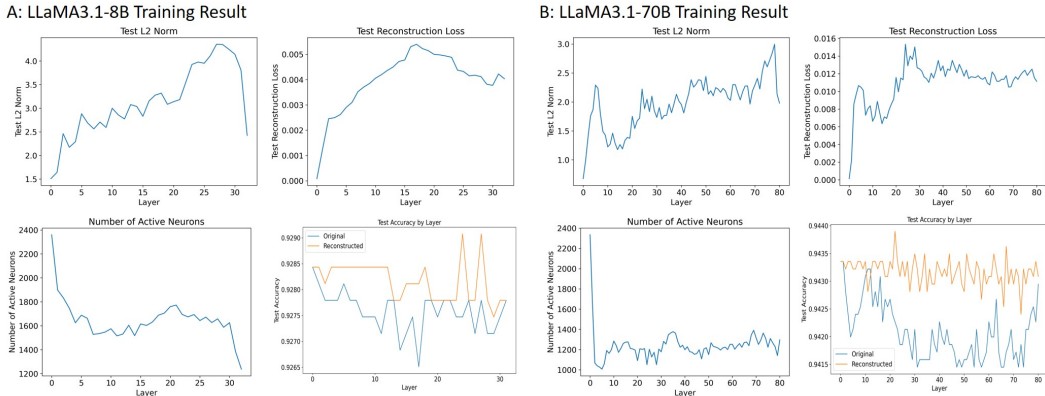

Figure 15: **Sparse Autoencoder (SAE) Training Metrics.** Each row represents different model architectures. From left to right, the panels illustrate the layer-wise test L2 norm, test reconstruction loss, and the number of active neurons during training. The top row corresponds to a smaller model (LLaMA3.1-8B), and the bottom row corresponds to a larger model (LLaMA3.1-70B) with more layers. Test Accuracy Between Original and Reconstructed Data. In both cases, reconstructed data achieves higher accuracy across layers, demonstrating the SAE's ability to preserve essential features during encoding and reconstruction.

# E   Replicated SAE Result in LLaMA3.1-8B

We investigated the role of the empowerment and uncertainty layers in LLaMA3.1-8B (temperature = 1) by training a Sparse Autoencoder (SAE) to identify and interpret their activations, followed by targeted interventions where each layer's activation was set to zero. The results, summarized in Figure 16A and Figure 16B, show that setting the empowerment layer to zero had a minimal effect on regression estimates and only slightly reduced model performance, suggesting that the empowerment layer has a limited role in sustaining task performance. In contrast, setting the uncertainty layer to zero led to a substantial reduction in regression estimates for uncertainty, accompanied by a marked decline in model performance. This highlights the critical importance of the uncertainty layer in facilitating exploration and maintaining robust decision-making capabilities within the model.

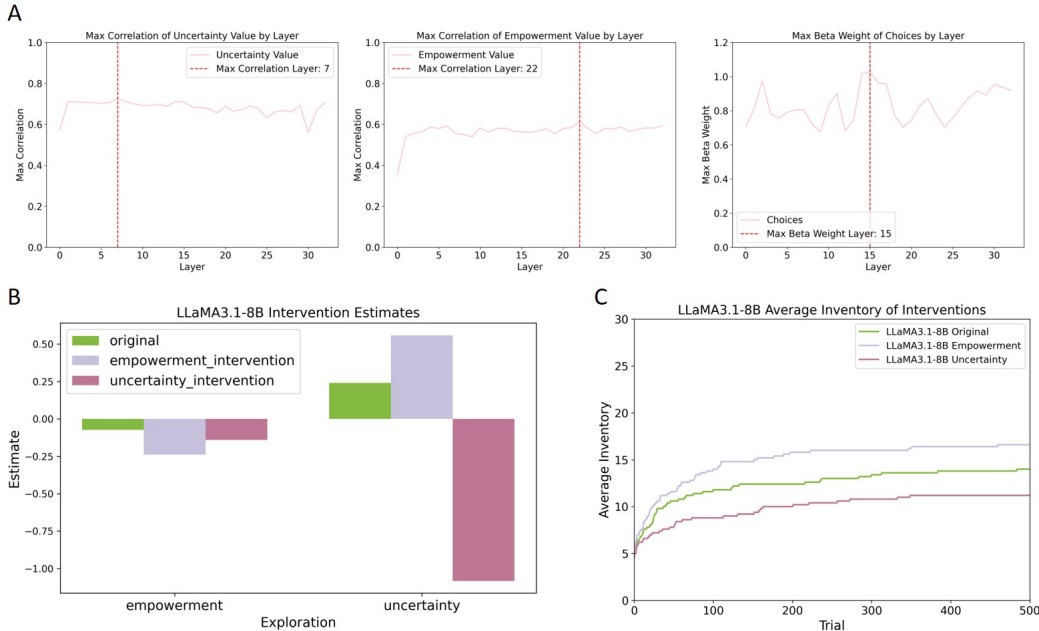

Figure 16: **A: SAE Correlation Analysis.** Maximum correlation of uncertainty values across layers, peaking at layer 7. Maximum correlation of empowerment values across layers, peaking at layer 22. Maximum beta weight of choices across layers, peaking at layer 15. **B: LLaMA3.1-8B Intervention Regression Results.** The bars represent the regression estimates for empowerment, and uncertainty under the original condition (LLaMA3.1-8B, temperature = 1), empowerment intervention (set to zero), and uncertainty intervention (set to zero). **C: LLaMA3.1-8B Average Inventory of Interventions.** Uncertainty intervention leads to a significant reduction in the average inventory, indicating its essential role in model performance.

