# OpenReview forum: "Large Language Models Think Too Fast To Explore Effectively"
_NeurIPS.cc/2025/Conference — NeurIPS 2025 poster_

### Official Review · Reviewer_V3R6 · 2025-06-05

**Clarity:** 4
**Significance:** 2
**Originality:** 2
**Rating:** 5
**Confidence:** 4

**Summary:**

This paper focuses on measuring the reasoning and creative capacity of LLMs by analyzing its performance on the Little Alchemy 2 game, which requires semantic understanding of combining elements.  It measures the performance of various LLMs on this task, defines measures for evaluation, and outlines results.

**Questions:**

While an interesting concept for the paper, LittleAlchemy 2 has been released for quite some time now and it is likely that the base models used have some of LittleAlchemy 2's data trained into their models.  A simple search on any search engine yields many pages with memorized combinations… which could have biased the LLM model when presented with preloaded training towards these questions.  How can we account for tainted base models?

How sensitive is this system to prompt variation?

In "past attempts" prompts, what is exactly included?  Is it just the suggestion of combinations or the suggestion paired with their observed outcome?  I expect that this will impact the outcomes.

**Ethical Concerns:**

["NO or VERY MINOR ethics concerns only"]

**Final Justification:**

Interesting and fun paper to read that is well written.  Adjusting score based on rebuttal arguments.  I am still a bit concerned about the limited domain scope and also have concerns about generalizability.  This potential lack of generalizability means that its significance may not be super high, but I appreciate the observations this paper provides and the arguments they pursue.

**Limitations:**

Yes

**Paper Formatting Concerns:**

Not aware of any major formatting issues

**Quality:**

3

**Strengths And Weaknesses:**

The experiments seem reasonably designed and assembled.  I appreciated the use of empowerment and uncertainty with a fixed testing regiment for the experimental design.  However, reasoning models and LLMs are two very different categories of capability and while paper delineates this, I didn't find calling out the differences between reasoning and non-reasoning models particularly useful towards the overall thesis that LLMs think too fast (especially when o1 clearly outperformed).  The observations don't feel particularly novel and align with what one would expect these systems to behave (e.g. with uncertainty).  However, it is useful to quantify these observations and experimental framework to aid future research that need to rely upon exploratory methods.  I would have appreciated more delineation based on the category of LLM (non-reasoning, reasoning) instead of individual callouts to specific models throughout the paper.

I appreciated the use of the empowerment and uncertainty metrics with a fixed budget of attempts allotted to each player as an experimental design.

While the comparison was made between DeepSeek and 4o with regards to usage of reasoning models, it would be curious to do the comparison instead against the top performing model of similar class (o1).  I really felt this part was pretty messy and should have clearly delineated the types of models by category more clearly.  In addition, be consistent throughout (it was unclear why Deepseek instead of o1 - which performed the best - was selected for a lot of the deep dive).

Some control for base model being tainted with answers from Little Alchemy 2 is likely necessary given the base models were probably tainted with many answers.

The observations are intriguing, but also are intuitive.  Doesn't feel like groundbreaking research, but it is perhaps a good validation to have this research affirm what is expected.

---

> ### Author Rebuttal · Authors · 2025-07-28
>
> We sincerely thank the reviewer for the thoughtful feedback and constructive suggestions. We appreciate the opportunity to clarify and refine our contributions in response to these insightful comments.
>
> ---
>
> ### On delineation between reasoning and non-reasoning models
>
> > “I would have appreciated more delineation based on the category of LLM (non-reasoning, reasoning) instead of individual callouts to specific models throughout the paper.”
>
> We appreciate this valuable suggestion. In our revision, we will add explicit labels in our figures and tables to clearly indicate which models are trained with intermediate reasoning traces (reasoning models) and which are not (non-reasoning models). This will improve clarity and provide a more principled framework for interpreting performance differences across model categories.
>
> ---
>
> ### On inconsistent use of reasoning model comparisons
>
> > “While the comparison was made between DeepSeek and 4o with regards to usage of reasoning models, it would be curious to do the comparison instead against the top performing model of similar class (o1). I really felt this part was pretty messy and should have clearly delineated the types of models by category more clearly. In addition, be consistent throughout (it was unclear why Deepseek instead of o1 - which performed the best - was selected for a lot of the deep dive).”
>
> Thank you for highlighting this issue. We acknowledge the inconsistency in using DeepSeek-R1 instead of o1 for certain deep analyses. At the time of our study, few publicly available reasoning models allowed access to reasoning traces. DeepSeek-R1 was chosen for its interpretability and accessibility, which enabled us to conduct detailed analyses such as thought diversity, empowerment tracking, and sparse representation.
>
> However, we agree that o1, being the top-performing model, should be more consistently highlighted. In the revised version, we will ensure that o1’s performance is included clearly in the results sections, while still using DeepSeek-R1 as the representative for thought-trace-level analysis due to its unique visibility into intermediate reasoning. We will also better delineate reasoning vs. non-reasoning categories throughout to prevent confusion.
>
> ---
>
> ### On contamination of base models with Little Alchemy 2 data
>
> > “LittleAlchemy 2 has been released for quite some time now and it is likely that the base models used have some of LittleAlchemy 2's data trained into their models... How can we account for tainted base models?”
>
> We acknowledge this concern and appreciate the opportunity to clarify. While it is possible that base models may have encountered some Little Alchemy 2 content during pretraining, we argue that contamination alone does not explain the observed behaviors for several reasons.
>
> First, model actions are constrained by the available inventory. Even if a model memorized a high-value combination (e.g., “human + metal”), it cannot execute it unless both elements are already present, which requires planning and multi-step reasoning.
>
> Second, successful gameplay in our task involves constructing long chains of dependent combinations, often exceeding 70 elements and 100+ steps. These long-horizon plans go beyond the retrieval of isolated facts and require flexible composition and foresight.
>
> Third, if contamination were the key driver, we would expect traditional large models (e.g., GPT-4o, LLaMA) to achieve similar performance. However, only reasoning-enhanced models (e.g., o1, DeepSeek-R1) succeed significantly, and only when they exhibit distinct strategic behaviors such as empowerment-driven exploration and thought diversity. This suggests that architecture and training objective, not mere memorization, are crucial.
>
> Furthermore, recent work by Jin et al. (2025) has shown a dissociation between factual memory and reasoning ability in large language models. LLMs may recall benchmark facts with high accuracy yet still fail to use them effectively in problem-solving, reinforcing our view that knowledge alone is insufficient.
>
> **Reference:**
> Jin, M., Luo, W., Cheng, S., Wang, X., Hua, W., Tang, R., ... & Zhang, Y. (2024). Disentangling memory and reasoning ability in large language models. *arXiv preprint* arXiv:2411.13504.
>
> ---
>
> ### On sensitivity to prompt variation
>
> > “How sensitive is this system to prompt variation?”
>
> We conducted prompt ablation and modification experiments, including variants with Chain-of-Thought prompting, explicit strategic guidance, and self-reflection (see Appendix C). These interventions did not significantly improve exploration performance. For example, the average inventory size remained around 43 (t = –1.17, p = 0.304) across prompts, suggesting that the system’s limitations are not primarily due to surface-level prompt phrasing.
>
> This insensitivity supports our hypothesis that deeper architectural factors—such as token-level autoregression and shallow context integration—underlie the fast-thinking bias in standard LLMs. In contrast, models trained with reasoning traces or fine-tuned on multi-step planning objectives show more robust improvement, suggesting that structural intervention is more effective than prompt engineering in this task.
>
> ---
>
> ### On the content of “past attempts” prompts
>
> > “In ‘past attempts’ prompts, what is exactly included? Is it just the suggestion of combinations or the suggestion paired with their observed outcome? I expect that this will impact the outcomes.”
>
> Thank you for this clarification request. The “past attempts” prompt includes both the attempted combinations and their corresponding outcomes. Specifically, it details whether each trial succeeded or failed, and in case of success, the resulting element. For example:
>
> Trial 0: water + fire → Success: 1, Result: steam.
> You successfully created a new element.
> Trial 1: steam + air → Success: 1, Result: mist.
> You successfully created a new element.
> Trial 2: water + earth → Success: 1, Result: mud.
> You successfully created a new element.
> Trial 3: mist + steam → Success: 0, Result: -1.
> Combining these two elements failed.
>
> This structure is designed to mirror the feedback available to human players in the real game. We agree with the reviewer that this information significantly affects the model’s behavior. Models that fail to incorporate failure feedback often repeat ineffective combinations, while models that learn from both successes and failures are more strategic and efficient in their exploration.
>
> We will clarify this aspect of the prompt in the final manuscript and explain its role in supporting learning from trial history.
>
> ---
>
> We again thank the reviewer for your thoughtful and constructive feedback. We hope our responses clarify our contributions and demonstrate the rigor of our experimental design and interpretation.

---

> > ### Comment · Reviewer_V3R6 · 2025-08-04
> >
> > I thank the authors for their thoughtful replies and clarifications.  I will raise my score.

---

### Official Review · Reviewer_cuDM · 2025-06-17

**Clarity:** 2
**Significance:** 2
**Originality:** 2
**Rating:** 3
**Confidence:** 3

**Summary:**

This paper investigates the ability of large language models (LLMs) to perform open-ended exploration, a critical cognitive capability for discovering new information and adapting to novel environments. Using the combinatorial puzzle game LittleAlchemy 2 , the authors compare how LLMs and humans explore new combinations of elements to create new ones. They find that most traditional LLMs underperform compared to humans in this task, primarily relying on uncertainty-driven strategies rather than empowerment-based reasoning—where actions are taken to maximize future possibilities.

**Questions:**

1. How do the findings apply to open-ended tasks beyond combinatorial games (e.g., autonomous scientific research or creative writing)? How generalizable are these findings beyond LittleAlchemy 2?

2. Can fine-tuning or prompting enhance the model's performance on such tasks or overcome the “fast thinking” limitation?

**Ethical Concerns:**

["NO or VERY MINOR ethics concerns only"]

**Final Justification:**

I would like to raise the score from 2 to 3. I still think that there should be some experiements exploring the potential of fine-tuning in such game tasks to improve LLMs' general reasoning abilities.

**Limitations:**

Yes

**Quality:**

2

**Strengths And Weaknesses:**

## Strengths

The paper studies a fundamental but understudied question: whether LLMs can engage in effective open-ended exploration akin to human cognition. The use of LittleAlchemy 2 as a benchmark for evaluating exploration strategies in LLMs is innovative and fills a gap in current evaluation frameworks. The integration of cognitive psychology concepts (empowerment, uncertainty) with deep learning analysis techniques (SAE) is unique and interdisciplinary. Its findings have implications for building more adaptive AI systems capable of discovery and innovation, especially in domains requiring long-term planning and intrinsic motivation.

## Weaknesses

1. It is better to demonstrate some illustrative figures for readers to understand this paper faster and clearer.

2. The findings are validated in Little Alchemy 2, but it remains unclear how they generalize to other open-ended tasks. The authors could discuss generalizability challenges or conduct more experiments in other domains.

3. Including more diverse models (e.g., open-source alternatives or smaller LLMs) would strengthen the robustness of conclusions.

4. The SAE analysis shows when empowerment and uncertainty are represented but lacks detailed mechanistic insights into why LLMs prioritize early-layer signals. Deeper analysis of transformer dynamics could enhance clarity. Besides, the SAE is only conducted on Llama3.1-70B. Including other reasoning models like QwQ or DeepSeek-Distilled-Qwen-32B can be more convincing.

---

> ### Author Rebuttal · Authors · 2025-07-28
>
> We thank the reviewer for the insightful feedback and constructive suggestions. The comments are guiding us toward better clarification and reflection on our contribution.
>
> ---
>
> ### On the need for illustrative figures
>
> > “It is better to demonstrate some illustrative figures for readers to understand this paper faster and clearer.”
>
> We fully agree with this suggestion. As our analyses span multiple levels, it would be beneficial to include an overview figure that summarizes the methods and findings to help readers navigate the content more efficiently. In the next revision, we plan to incorporate such a figure (e.g., a schematic pipeline) to enhance readability and support intuitive understanding of our approach.
>
> ---
>
> ### On generalizability beyond Little Alchemy 2
>
> > “The findings are validated in Little Alchemy 2, but it remains unclear how they generalize to other open-ended tasks. The authors could discuss generalizability challenges or conduct more experiments in other domains.”
>
> We appreciate the reviewer’s thoughtful concern. We agree that extending the findings to other open-ended domains, such as scientific discovery or creative writing, would enhance the generalizability of our claims. However, the primary constraint is the limited availability of large-scale human data in open-ended exploration environments. Little Alchemy 2 offers a rare combination of complex combinatorial structure and large-scale behavioral data.
>
> Furthermore, our current analysis is closely tied to cognitive mechanisms unique to this task. The latent model, sparse autoencoder, and thought analyses all rely on this task’s affordances. While this limits breadth, it allows us to conduct a much deeper investigation. We view this as a deliberate trade-off between generalization and insight. We hope to explore additional tasks in future work that test the same core mechanisms.
>
> ---
>
> ### On model diversity and transformer dynamics
>
> > “Including more diverse models (e.g., open-source alternatives or smaller LLMs) would strengthen the robustness of conclusions.
> > The SAE analysis shows when empowerment and uncertainty are represented but lacks detailed mechanistic insights into why LLMs prioritize early-layer signals. Deeper analysis of transformer dynamics could enhance clarity. Besides, the SAE is only conducted on Llama3.1-70B. Including other reasoning models like QwQ or DeepSeek-Distilled-Qwen-32B can be more convincing.”
>
> Thank you for this suggestion. To address this concern, we implemented additional experiments with Qwen-2.5-32B and QwQ-32B. Our results show that Qwen-2.5-32B discovered 14 elements on average, which is lower than the human average of 42. In contrast, QwQ-32B discovered 48 elements on average, outperforming humans. While we did not have sufficient time during the rebuttal phase to replicate the full SAE and thought analysis for these models, the raw exploration results demonstrate that our core findings generalize beyond LLaMA and GPT-style models.
>
> Specifically, QwQ is a benchmark reasoning model in the Qwen series, directly trained using reinforcement learning on top of Qwen-2.5-32B. These results suggest that with similar architecture and pretraining, reasoning-enhanced variants significantly improve exploration performance. This reinforces our central claim that stronger reasoning capabilities contribute to better open-ended exploration. Additionally, these results align with our earlier finding that model size positively correlates with exploration quality. Both Qwen variants’ performances are consistent with their parameter scale. We will be happy to include full SAE and thought-level analyses of these models in a future version.
>
> | Model               | Estimated Empowerment | Estimated Uncertainty |
> |---------------------|------------------------|------------------------|
> | **Qwen-2.5-32B**                | **-0.01777**               | **-1.35386**               |
> | **QwQ-32B**             | **0.088587**               | **2.752409**               |
> | LLaMA3-8B           | -0.05991               | -1.25665               |
> | LLaMA3-70B          | -0.00409               | -1.42584               |
> | GPT-4o              | -0.16928               | 0.249572               |
> | DeepSeek-R1  | 0.140074               | 4.181993               |
> | o1                  | 0.722223               | 4.151111               |
> | Human          | 0.378              | 0.223             |
>
> ---
>
> ### On generalization to broader open-ended domains
>
> > “How do the findings apply to open-ended tasks beyond combinatorial games (e.g., autonomous scientific research or creative writing)? How generalizable are these findings beyond LittleAlchemy 2?”
>
> We appreciate the opportunity to further reflect on the generalizability of our study. While our current work focuses on a well-controlled yet open-ended environment, we believe the underlying insight—namely, that reasoning capacity enhances exploration—has broader relevance. We identified one concurrent work by Grams et al. (2025), which evaluated exploration in LLMs in a maze-like environment. They found that (1) exploration improves with model scale and (2) prompting for reasoning significantly enhances exploratory behavior. Both findings are consistent with our conclusions.
>
> These results collectively point to a shared principle across domains: reasoning encourages goal-directed discovery and strategic coverage of action space. While task details differ, we believe the cognitive basis of exploration is robust and worthy of further empirical study across environments.
>
> **Reference:**
> Grams, T., Betz, P., & Bartelt, C. (2025). Disentangling exploration of large language models by optimal exploitation. *arXiv preprint* arXiv:2501.08925.
>
> ---
>
> ### On enhancing exploration via fine-tuning and prompting
>
> > “Can fine-tuning or prompting enhance the model's performance on such tasks or overcome the 'fast thinking' limitation?”
>
> Thank you for this excellent question. We believe fine-tuning with reasoning data, particularly through supervised or reinforcement learning, can enhance the model’s exploration ability. However, we caution against directly fine-tuning on our specific task, as this could lead to data leakage. The essence of open-ended exploration lies in learning the environment’s structure from scratch, rather than memorizing specific combinations.
>
> That said, models trained on general reasoning tasks (e.g., o1, DeepSeek Reasoner) do generalize well to our domain. This suggests that pretraining on reasoning-related tasks enhances the model’s ability to explore unseen environments. We also tested prompt engineering with traditional models, but the task appears too complex for prompting alone to be effective. Results can be found in Appendix C.
>
> ---
>
> We once again thank the reviewer for your thoughtful and constructive comments. We hope our responses address the questions and demonstrate the strength and rigor of our study.

---

> > ### Comment · Reviewer_cuDM · 2025-08-02
> > **Reply to the rebuttal**
> >
> > I thank the authors for their rebuttal. I still have some questions on enhancing exploration via fine-tuning. Some prior works have shown that fine-tuning LLMs using self-playing on some games can help improve the general reasoning ability. If we cannot apply fine-tuning to improve the LLMs' abilities on such a puzzle game, can we leverage the data generated in this game to fine-tune LLMs to improve their general reasoning abilities (in the domains such as math, science)?
> >
> > [1] Cheng, Pengyu, et al. "Self-playing adversarial language game enhances llm reasoning." Advances in Neural Information Processing Systems 37 (2024): 126515-126543.
> >
> > [2] Zhao, Andrew, et al. "Absolute zero: Reinforced self-play reasoning with zero data." arXiv preprint arXiv:2505.03335 (2025).

---

> > > ### Author Response · Authors · 2025-08-02
> > >
> > > Thanks to the reviewer for the engaging discussion. We agree that fine-tuning, particularly with reinforcement learning (RL) algorithms, can help improve model performance on our task. The reasoning models tested in the paper, as well as our additional results, already suggest that models fine-tuned with general RL algorithms can perform very well—sometimes even surpassing human performance.
> > >
> > > In our rebuttal, we emphasize that models could indeed perform better on our task by improving their general reasoning capacity. However, we do not recommend directly fine-tuning models on our task itself, as this raises concerns about data leakage. As researchers focused on exploration, we prioritize the generalization ability of models over task-specific optimization.
> > >
> > > We also appreciate the reviewer’s follow-up question. It is indeed interesting to consider whether training on our task could in turn help models generalize better to real-world applications. While we currently lack empirical evidence, we believe this is a promising direction. For instance, systems like *AlphaEvolve* (Novikov et al., 2025) leverage the in-context learning abilities of large language models to "search" for novel solutions in domains such as mathematics and scientific discovery. This kind of search is driven by reward signals and often involves combinatorial reasoning—much like our task setup.
> > >
> > > If models were augmented with mechanisms like *empowerment*-driven reasoning, as in our approach, they might explore solution spaces more efficiently by reasoning about the underlying semantic properties of their components. We believe this could directly benefit domains such as algorithm discovery and scientific inference.
> > >
> > > Again, we thank the reviewer for raising these thought-provoking points. We would be happy to incorporate this discussion into our revised manuscript to further highlight the broader impact of our work.
> > >
> > > **References**:
> > > Novikov, A., Vũ, N., Eisenberger, M., Dupont, E., Huang, P. S., Wagner, A. Z., ... & Balog, M. (2025). *AlphaEvolve: A coding agent for scientific and algorithmic discovery*. arXiv preprint arXiv:2506.13131.
> > > Castro, P. S., Tomasev, N., Anand, A., Sharma, N., Mohanta, R., Dev, A., ... & Stachenfeld, K. L. (2025). *Discovering symbolic cognitive models from human and animal behavior*. bioRxiv, 2025-02.

---

### Official Review · Reviewer_Envx · 2025-06-22

**Clarity:** 2
**Significance:** 3
**Originality:** 2
**Rating:** 4
**Confidence:** 4

**Summary:**

The authors investigate how well the current state-of-the-art LLMs (LLaMa, GPT, o1, DeepSeek) perform in an open-ended, novel discovery task when compared to humans. Their extensive experiment shows that, when compared to humans, LLaMa models significantly underperform, GPT4 performs somewhat comparably but doesn't excel, and DeepSeek and o1 surpass human level performance. The authors attribute the underperformance of many LLMs to their heavy reliance on uncertainty driven heuristics and rash decisions, rather than incorporating empowerment, or "the intrinsic desire to create elements that offer many more" future discoveries. Their deep-dive with regression and SAE shows that uncertainty-based processing happens in early transformer layers, while empowerment signals only emerge later—by which time the model has often already made a premature decision.

**Questions:**

1. In section 4.1, the reported results show that DeepSeek also vastly outperformed humans in Little Alchemy 2. Is there a reason why o1 is highlighted, and less focus is put on the achievement of DeepSeek? Later in 4.3, I see that the comparison is made between GPT and DeepSeek, not with o1. I presume that this deep dive in 4.3 was easier with DeepSeek than with o1 because the former is open source while the latter is proprietary. Given this, wouldn’t it have been helpful for consistency and completeness to also emphasize DeepSeek’s strong performance more prominently in section 4.1?
2. In section 4.4, the authors provide a detailed analysis of exploration behavior and decision dynamics specifically for LLaMA models. Is there a particular reason why a similar in-depth examination wasn’t conducted for GPT or DeepSeek models? I think it is critical that the authors demonstrate whether the patterns observed with LLaMA generalize across other model families.

**Ethical Concerns:**

["NO or VERY MINOR ethics concerns only"]

**Final Justification:**

I thank the authors for their detailed clarifications and responses to my concerns. After some consideration, I decided to keep my rating of 4, because I still believe that a more level comparisons between different family of models is critical for the completion of the authors' reaserch.

**Limitations:**

Yes

**Paper Formatting Concerns:**

No major formatting concerns, but a few grammatical errors and stylistic nit:

1. Lines 139-140: "Selecting combinations that maximize future potential ([...])." is a fragmented sentence.
2. Lines 161-162: "If the combination is repeated the successful combinations and no new elements are created, [...]" doesn't make sense. Was the intention to say "If a successful combination is repeated and no new elements are created, [...]"?
3. Line 226: Is "discouvered" a type for "discovered"?
4. Authors often start sentences with "This" as a pronoun, which can be ambiguous. They should err on the side of using it as a determiner-- e.g. "This observation", "This example"--to make the reference clearer.

**Quality:**

3

**Strengths And Weaknesses:**

It is no trivial work to perform such a comprehensive study across multiple LLMs, and to conduct an impressively large scale human experiment for providing a robust and meaningful benchmark. The paper also offers a novel diagnosis that LLMs tend to "think too fast" to explain why they often underperform on open-ended, exploratory tasks than humans. Section 4, in particular, is well written: it starts with a clear summary of results that sets the town for the rest of the section, then methodically delves into the mechanisms behind why LLMs exhibit premature decision-making behavior.

Meanwhile, I believe the authors could've made more level comparisons between different family of models. The strongest empirical results hinges on o1, whose technical details are unknown. DeepSeek, a strong runner-up with impressive performance and transparency, deserved greater emphasis for better interpretability and reproducibility. The fact that different model families in separate sections made it somewhat challenging to grasp how the authors' conclusions broadly apply across all LLMs.

---

> ### Author Rebuttal · Authors · 2025-07-28
>
> We thank the reviewer for the insightful reviews and constructive feedback. Below we address your main concerns and questions in detail.
>
> ---
>
> ### On the inconsistent emphasis on o1 and DeepSeek-R1 across analyses
>
> > “In section 4.1, the reported results show that DeepSeek also vastly outperformed humans in Little Alchemy 2. Is there a reason why o1 is highlighted, and less focus is put on the achievement of DeepSeek? Later in 4.3, I see that the comparison is made between GPT and DeepSeek, not with o1. I presume that this deep dive in 4.3 was easier with DeepSeek than with o1 because the former is open source while the latter is proprietary. Given this, wouldn’t it have been helpful for consistency and completeness to also emphasize DeepSeek’s strong performance more prominently in section 4.1?”
>
> Thank you for pointing this out—we realize that the current presentation may have created confusion for the reader. When conducting a mechanistic analysis, we must ensure the models are transparent and feasible to deploy for the required analyses. As you correctly noted, while o1 demonstrates strong performance, its training details and model architecture are not publicly available. In contrast, DeepSeek-R1 is open-sourced and allows deeper analysis of its internal behavior.
>
> Therefore, although we highlighted o1's top performance in Section 4.1, the chain-of-thought outputs and internal activations that we could analyze in detail were only available for DeepSeek-R1. This discrepancy resulted in some imbalance across sections. We appreciate your suggestion and, to improve clarity and consistency, we will revise the paper to give more consistent emphasis to DeepSeek-R1’s performance and include more explicit justification for why certain models were selected for in-depth analysis. We will still report o1’s results due to their significance but shift the highlight to DeepSeek-R1 to better align with the scope of our analyses. Thank you again for your helpful comment.
>
> ---
>
> ### On why SAE analysis was only conducted for LLaMA models
>
> > “In section 4.4, the authors provide a detailed analysis of exploration behavior and decision dynamics specifically for LLaMA models. Is there a particular reason why a similar in-depth examination wasn’t conducted for GPT or DeepSeek models?”
>
> This is a great point, and the reason is closely related to the one above. Our SAE analysis requires access to the complete hidden activations of all transformer layers to train a separate sparse autoencoder model. This analysis is only feasible on models that are fully open-sourced and reasonably deployable.
>
> GPT-4o and o1 are not open-source, so we do not have access to their internal activations. While DeepSeek-R1 is open-source, the model has more than 600 billion parameters, which makes it extremely challenging to deploy for this purpose. Obtaining full layer activations across multiple inputs would require substantial hardware—far beyond our computational resources. For reference, the activation size would be at least several times larger than that of LLaMA3.1-70B, which was already the upper limit of what we could analyze.
>
> Given these constraints, we chose LLaMA models for this layerwise analysis because they strike a practical balance between openness, performance, and computational accessibility. We hope this helps clarify the decision.
>
> ---
>
> ### On potential alternative models for future analysis
>
> While we could not analyze DeepSeek-R1 with SAE at the time of this study, we acknowledge that the availability of more lightweight, transparent reasoning models has improved. In particular, newer models like the Qwen series and its reasoning-optimized variant QwQ-32B are promising. QwQ is trained via reinforcement learning on top of Qwen-2.5-32B to enhance reasoning capabilities and is released with sufficient transparency for deeper investigation.
>
> If we had access to these models earlier, we would have incorporated parallel SAE and thought-level analyses between Qwen-2.5-32B and QwQ-32B. We refer you to our response to Reviewer cuDM for a discussion of results using these models. We hope this gives a more straightforward illustration of the difference between traditional LLMs and reasoning-enhanced models in both reasoning and exploration.
>
> ---
>
> ### On formatting and grammar concerns
>
> > “No major formatting concerns, but a few grammatical errors and stylistic nit: [...]”
>
> Thank you for the careful reading and suggestions regarding grammar and clarity. We will correct the specific sentence fragments, typos (e.g., “discouvered”), and ambiguous uses of pronouns (e.g., replacing “this” with more specific noun phrases). We will also review the manuscript more broadly to fix similar issues throughout. We appreciate your attention to detail and will make sure the final version reflects these improvements.
>
> ---

---

### Official Review · Reviewer_BscT · 2025-06-24

**Clarity:** 3
**Significance:** 3
**Originality:** 4
**Rating:** 5
**Confidence:** 3

**Summary:**

This submission studies the exploration abilities of LLMs in open-ended tasks using the game Little Alchemy 2, a combinatorial task where the goal is to combine existing elements in order to discover as many new elements as possible. The authors evaluate the perfomance of several LLMs, specifically GPT-4o, GPT-o1, DeepSeek-R1, LLaMA3.1-8B, and LLaMA3.1-70B, on this task and compare their performance to human players. They find that the LLMs underperform on this task relative to humans, with the exception of GPT-o1 and DeepSeek-R1.

The authors distinguish between uncertainty-driven exploration, a common paradigm in RL and bandits where the learner takes acions favoring novelty, and empowerment-based exploration, a concept from human learning that favors taking actions (in this case, combinations of elements) thay lead to more downstream possibilities. They show that in LLaMA, uncertainty signals dominate early transformer layers and heavily influence decisions, while empowerment signals arise later in the network and are often unused. Thus, LLMs “thinking too fast” is one explanation for why most models achieve sub-human level performance on this task.

**Questions:**

How does uncertainty-based exploration compare to empowerment-based exploration? Are there settings in which one is useful but not the other?

What other types of tasks (besides Little Alchemy 2) do you believe your findings generalize to?

**Ethical Concerns:**

["NO or VERY MINOR ethics concerns only"]

**Final Justification:**

I believe this paper clears the bar for NeurIPS due to the authors' fresh take on the problem of using LLMs for exploration in decision making tasks. While the authors are not the first to look at such questions, they are the first (to my knowledge) to take a human learning-based perspective, as previous work has predominantly focused on uncertainty-driven explanation. The use of Little Alchemy 2 provides a natural task to test exploration abilities of large models, and the authors’ methodology for measuring empowerment is well-grounded in prior work.

**Limitations:**

Yes.

**Quality:**

3

**Strengths And Weaknesses:**

Strengths:

This paper adds to the quickly growing literature on using LLMs for exploration in decision making tasks. While the authors are not the first to look at such questions, they are the first (to my knowledge) to take a human learning-based perspective, as previous work has predominantly focused on uncertainty-driven explanation. The use of Little Alchemy 2 provides a natural task to test exploration abilities of large models, and the authors’ methodology for measuring empowerment is well-grounded in prior work.

I particularly liked the authors’ use of sparse autoencoders fo rrepresentational analysis, which led to some interesting finding (see above in summary).

More generally, I thought that the experiments were well thought out, and the results made sense and are reproducible. I also found the paper to be well-written.


Weaknesses:

The parts of the paper that were most interesting to me were the claims about models “thinking too fast”. However, it would have been nice if more time was dedicated to formalizing this notion a bit more precisely.

While the empowerment metric is well-motivated, the reliance on an external neural network to compute empowerment scores (as opposed to using some sort of ground-truth signal) is somewhat undesirable.

I think that this paper could benefit from a more thorough comparision with the literature on using LLMs to make decisions under uncertainty. While I believe the authors that most of this work has focused on uncertainty-based exploration, there is still enough overlap in some of these works that they are worth mentioning and comparing against. I've included an (incomplete) list of such papers below, for your reference.

Toward Efficient Exploration by Large Language Model Agents by Dilip Arumugam and Thomas L. Griffiths

CAN FOUNDATION MODELS ACTIVELY GATHER INFORMATION IN INTERACTIVE ENVIRONMENTS TO TEST HYPOTHESES? by Ke et al

Beyond Numeric Rewards: In-Context Dueling Bandits with LLM Agents by Xia et al

Learning to Explore: An In-Context Learning Approach for Pure Exploration by Russo et al

Should You Use Your Large Language Model to Explore or Exploit? by Keegan Harris and Aleksandrs Slivkins

Do llm agents have regret? a case study in online learning and games by Park et al

---

> ### Author Rebuttal · Authors · 2025-07-28
>
> We thank the reviewer for the insightful feedback and constructive suggestions. The comments are guiding us toward better clarification and reflection on our contribution.
>
> ---
>
> ### On the use of a neural network instead of ground-truth for empowerment
>
> > “While the empowerment metric is well-motivated, the reliance on an external neural network to compute empowerment scores (as opposed to using some sort of ground-truth signal) is somewhat undesirable.”
>
> We appreciate the reviewer’s observation on this detail. Indeed, in our experiment—as well as in previous studies of human exploration—using ground truth would be more straightforward and simpler. However, using ground truth necessarily assumes that participants and agents already know the complete rule structure of the task environment, which is in fact the implicit goal of this task: to explore and uncover the “rules of the world” and exploit them.
>
> The neural network is trained only on partial ground truth, using word2vec embeddings of each element, aiming to learn a latent world model from the game's observable patterns. Through its predictions, we are simulating how general semantic meaning can be used to estimate empowerment in this task. This setup better reflects how a cognitive agent might operate under uncertainty with incomplete knowledge.
>
> ---
>
> ### On the concept of “thinking too fast” and its formalization
>
> > “The parts of the paper that were most interesting to me were the claims about models ‘thinking too fast’. However, it would have been nice if more time was dedicated to formalizing this notion a bit more precisely.”
>
> We truly agree with the reviewer’s comment here. It is indeed challenging to formally define what “thinking too fast” means in the context of LLMs. Our intuition comes from how human thinking speed is typically measured—such as through response time in cognitive tasks. Meanwhile, when we talk about thinking in general, “fast” may imply shallow, intuitive reasoning, as discussed in Kahneman’s *Thinking, Fast and Slow* (2011).
>
> Although LLMs do not have response times in the same sense as humans, transformer processing follows a strict temporal structure: a single forward pass is processed sequentially from the first layer to the last. This gives rise to a notion of “early” versus “late” processing. Our SAE analysis shows that uncertainty-driven exploration is mostly represented in early layers, whereas empowerment emerges in later layers. This creates a natural analogy to fast (shallow) versus slow (deep) thinking, similar to how response time correlates with cognitive depth in humans.
>
> This analogy is further supported by Hu et al. (2025), who demonstrate that transformer layerwise processing exhibits reaction-time-like signatures: early layers correspond to fast, low-effort responses, while deeper layers reflect more accurate, reflective reasoning—closely paralleling human response time and accuracy tradeoffs.
>
> Additionally, our thought analysis revealed that both the computational resources consumed and the structure of the generated thoughts suggest that reasoning models spend more tokens in “thinking” and engage in more comprehensive mental exploration in the conceptual space—closely resembling System 2 processing. This is also consistent with findings by de Varda et al. (2025), who show that reasoning-capable LLMs incur cognitive costs analogous to humans when engaging in deeper reasoning.
>
> Therefore, while our current notion of “thinking too fast” is not yet formally defined, our analogy—grounded in cognitive theory and supported by computational evidence—offers a meaningful and interpretable perspective. We will take this seriously in future work and aim to develop a more systematic, formal framework to quantify this phenomenon in LLMs.
>
> ---
>
> ### On related literature regarding uncertainty-based decision-making
>
> > “I think that this paper could benefit from a more thorough comparison with the literature on using LLMs to make decisions under uncertainty.”
>
> Thank you for the supplemented references. We recognize that some of these works are highly relevant and valuable. We would be happy to include them in the related work and discussion sections to strengthen our literature background in the next revision. These works help situate our approach relative to uncertainty-based and in-context exploration strategies, which we complement by highlighting empowerment as an additional, semantically grounded principle of exploration.
>
> ---
>
> ### On the comparison between uncertainty-based and empowerment-based exploration
>
> > “How does uncertainty-based exploration compare to empowerment-based exploration? Are there settings in which one is useful but not the other?”
>
> Uncertainty-driven exploration in this task specifically means agents look back at their history and determine which elements have been selected more and which less. Elements selected less frequently may contain more unknowns and therefore higher uncertainty about their utility in the task, which draws the agent's attention.
>
> Empowerment, by contrast, is more semantically driven. It involves estimating, based on the meaning of the words (concepts), how fruitful a given element could be in generating new combinations. This potentially leads to a higher chance of discovery. Thus, empowerment requires a deeper understanding of the world—effectively a learned world model.
>
> As shown in the Appendix A, once more elements are discovered (meaning the decision space is expanding), relying solely on uncertainty becomes less effective: with so many possibilities, many of them unproductive, sampling under uncertainty alone is inefficient. To explore effectively, an agent must use its semantic knowledge and internal world model to selectively pick out the elements most worth trying, which demands deeper reasoning. Therefore, this task naturally favors empowerment-driven over uncertainty-driven exploration, especially within a limited trial budget.
>
> ---
>
> ### On generalization to other task domains
>
> > “What other types of tasks (besides Little Alchemy 2) do you believe your findings generalize to?”
>
> The first and most direct type is compositional tasks. Since this task involves compositionality, learning, and planning, we believe the heuristics identified here in both humans and AI models are widely used in other compositional domains as well.
>
> More generally, our findings should apply to tasks that require internal world models and prior knowledge, rather than learning entirely from scratch. This Little Alchemy 2 task implicitly requires such prior knowledge: for instance, if an agent does not understand the meaning of “water” or “fire” in real-world terms, it becomes nearly impossible to simulate the outcome of combining them (e.g., creating “steam”) in its internal reasoning.
>
> These kinds of tasks are harder to measure but are increasingly receiving attention. The need for prior semantic grounding and compositional generalization is a common thread across many such domains.
>
> ---
>
> **References**
>
> - Hu, J., Lepori, M. A., & Franke, M. (2025). *Signatures of human-like processing in Transformer forward passes*. arXiv preprint arXiv:2504.14107.
> - Kahneman, D. (2011). *Thinking, fast and slow*. Macmillan.
> - de Varda, A., D’Elia, F. P., Fedorenko, E., & Lampinen, A. (2025, July 24). *The cost of thinking is similar between large reasoning models and humans*. OSF Preprints. https://doi.org/10.31234/osf.io/m2cu5_v1

---

> > ### Comment · Reviewer_BscT · 2025-08-02
> >
> > Thanks for your reply. I will maintain my score, and will engage with the other reviewers throughout the rest of the discussion period.

---

### Decision · Program_Chairs · 2025-09-17

**Decision:**

Accept (poster)

**Comment:**

This paper studies the question of whether current LLMs can explore effectively in open-ended settings. Focusing on the Little Alchemy 2 task, the authors find that standard models underperform human baselines, while reasoning-enhanced models (e.g., o1, DeepSeek-R1) are competitive or superior. To understand this, the authors distinguish between "uncertainty-driven" and "empowerment-driven strategies", and perform an analysis based on SAEs which suggests that uncertainty is represented in early blocks while empowerment emerges in later blocks, leading to “thinking too fast” and premature choices.

Reviewers generally felt that the paper addresses a timely and interesting problem, and is well written. Many of the initial issues raised by the reviewers were addressed in the discussion period (e.g. corroborating results (Qwen-2.5-32B vs. QwQ-32B) that align with the central claim, clarified prompt design/insensitivity). While the question of expanding to domains beyond Little Alchemy 2 remains for future work (the main concern remaining after discussion), I believe the central contribution clears the bar for NeurIPS. For the camera ready, I encourage the authors to include the interesting many points that came up in discussions with reviewers.